# Effects of different water quality regulators on growth performance, immunologic function, and domestic water quality of GIFT tilapia

Liang-Gang Wang[1], Meng-Qian Liu[1], Xiao-Dong Xie[1], Yu-Bo Sun[1], Ming-Lin Zhang[1], Yi Zhao[1], Qi Chen[1], Yi-Qu Ding[1], Mei-Ling Yu[1], Zheng-Min Liang[1,2], Ting-Jun Hu[1,2], Wan-Wen Liang[3]*, Ying-Yi Wei[1,2]*

1 College of Animal Science and Technology, Guangxi University, Nanning, PR China, 2 Guangxi Zhuang Autonomous Region Engineering Research Center of Veterinary Biologics, Nanning, PR China, 3 Guangxi Key Laboratory of Aquatic Genetic Breeding and Healthy Aquaculture, Guangxi Academy of Fishery Sciences, Nanning, PR China

* weiyingyi@gxu.edu.cn (YYW); nnlww@126.com (WWL)

**Data Availability Statement:** All relevant data are within the paper and its Supporting Information files.

## Abstract

Water quality regulation is widely recognized as a highly effective strategy for disease prevention in the field of aquaculture, and it holds significant potential for the development of sustainable aquaculture. Herein, four water quality regulators, including potassium monopersulfate (KMPS), tetrakis hydroxymethyl phosphonium sulfate (THPS), *bacillus subtilis* (BS), and chitosan (CS), were added to the culture water of *Oreochromis niloticus* (GIFT tilapia) every seven days. Subsequently, the effects of these four water quality regulators on GIFT tilapia were comprehensively evaluated by measuring the water quality index of daily growth-related performance and immune indexes of GIFT tilapia. The findings indicated that implementing the four water quality regulators resulted in a decrease in the content of ammonia nitrogen, active phosphate, nitrite, total organic carbon (TOC), and chemical oxygen demand (COD) in the water. Additionally, these regulators were found to maintain dissolved oxygen (DO) levels and pH of the water effectively. Furthermore, using these regulators demonstrated positive effects on various physiological parameters of GIFT tilapia, including improvements in final body weight, weight gain rate (WGR), specific growth rate (SGR), condition factor (CF), feed conversion ratio (FCR), spleen index (SI), hepatosomatic index (HSI), immune cell count, the activity of antioxidant-related enzymes (Nitric oxide, NO and Superoxide dismutase, SOD), and mRNA expression levels of immunity-related factors (Tumor Necrosis Factor-alpha, TNF-α and Interleukin-1 beta, IL-1β) in the liver and spleen. Notably, the most significant improvements were observed in the groups treated with the BS and CS water quality regulators. Moreover, BS and CS groups exhibited significantly higher serum levels of albumin (ALB) and total protein (TP) ($P < 0.05$), whereas the other indicators showed no significant difference ($P > 0.05$) compared to the control group. However, the KMPS and THPS groups of GIFT tilapia exhibited significantly higher serum levels of aspartate aminotransferase (AST), alanine transaminase (ALT), creatinine (CRE) and blood urea nitrogen (BUN) ($P < 0.05$), whereas they exhibited significantly decreased HSI ($P < 0.05$). In addition, the partially pathological observations revealed the

**Funding:** This work was financially supported by the Innovation Driven Development Fund of Guangxi [Grant number: GK AA17204081-2]. The Guangxi innovation team building project of the national modern agricultural industry technology system [Grant number: nycytxgxcxtd-14-02].We thank Guangxi University for its resources and support for this research result. The funders had no role in study design, data collection and analysis, decision to publish, or preparation of the manuscript.

**Competing interests:** The study does not involve conflicts of interest on any front.

presence of cell vacuolation, nuclear shrinkage, and pyknosis within the liver. In conclusion, these four water quality regulators, mainly BS and CS, could improve the growth performance and immunity of GIFT tilapia to varying degrees by regulating the water quality and then further increasing the expression levels of immune-related factors or the activity of antioxidant-related enzymes of GIFT tilapia. On the contrary, the prolonged use of KMPS and THPS may gradually diminish their growth-enhancing properties and potentially hinder the growth of GIFT tilapia.

## 1. Introduction

The GIFT strain of tilapia (*Oreochromis niloticus*) possesses several advantageous traits, including fast growth, miscellaneous diet, hypoxia tolerance, high yield, good meat quality, and strong adaptability [1]. The prevalence of tilapia diseases is exacerbated by environmental pollution and germplasm, which significantly reduce the meat quality of tilapia and bring huge losses to the tilapia culture industry. Therefore, it is crucial to prioritize improving the culture environment or preventing diseases. In addition to using drugs to enhance the immunity of aquatic animals aiming at reducing disease occurrence, another important method to prevent diseases is by regulating water quality to improve the living environment of these animals. In the intensive culture of tilapia, various substances such as residual bait, feces, dead algae, and microorganisms accumulate at the bottom of the pond. As the culture time extends, these substances decompose and give rise to toxic substances like ammonia, nitrogen, and nitrite. Unfortunately, this pollution greatly affects the water body and hampers the growth of tilapia. In severe cases, it can even lead to the death of the fish [2, 3]. Herin, we employed four different types of water quality regulators, including Potassium monopersulfate (KMPS), tetrakis hydroxymethyl phosphonium sulfate (THPS), *bacillus subtilis* (BS), and chitosan (CS). In brief, KMPS occurs as a series of chain reactions after dissolving in water, resulting in various reactive oxygen species and free radicals. These active substances can destroy the permeability of microbial cell membranes, thus interfering with the DNA and RNA synthesis of pathogens, thereby showing significant germicidal and inactivating effects [4]. Moreover, KMPS has a good bacteriostatic and algae-killing effect, in addition to being of great significance in sludge degradation and sediment improvement [5]. The water quality regulator known as THPS is a highly effective sediment enhancer that exhibits potent bactericidal and algicidal properties, while also facilitating the decomposition of residual organic matter such as bait and fecal material present at the bottom of ponds [6]. The BS can promote growth by secreting various digestive enzymes [7, 8] and improving intestinal flora stability and tilapia immunity [9]. In addition, BS can quickly decompose food, feces, and other residual organic matter in the water. BS also can slow down water pollution, improve water microflora and tilapia growth performance, and even immune function [10]. The CS is a cationic polysaccharide used widely in many fields. For instance, CS is often used as a feed additive owing to possessing multiple characteristics, including bacteriostatic, antiviral, antioxidant, cholesterol-lowering, growth-promoting, and immunity-enhancing effects [11]. Additionally, CS is widely used in aquaculture due to its good biosafety, biodegradability, and biocompatibility [12, 13]. Moreover, CS can effectively degrade nitrite, phosphate, and ammonia nitrogen contents in aquatic wastewater [14, 15], besides having a continuous adsorption effect on pollutants in water [16]. Currently, aquaculture predominantly relies on pharmaceutical interventions for disease prevention and treatment, thereby significantly augmenting the economic burden associated with aquaculture operations. Prolonged administration of drugs not only leads to the

accumulation of drug residues but also fosters the development of drug-resistant pathogenic bacteria [17]. Consequently, this study aims to comprehensively evaluate the effects of KMPS, THPS, BS and CS on GIFT tilapia by determining the growth performance, blood physiology and biochemistry, histopathology, antioxidant enzymatic activity, and immunity-related factors. Eventually, our results may provide a theoretical basis for selecting and using the best effective water quality regulators in aquaculture.

## 2. Materials and methods

### 2.1. Ethics code of conduct

Throughout this experiment, we strictly adhered to the ethical principles of the China Experimental Animal Welfare Ethics Committee.

### 2.2. Experimental animal

Totally, 1500 GIFT tilapia weighing 475.50 ± 4.05 g were collected from the National Guangxi Nanning Tilapia Seed Farm of Guangxi Fisheries Research Institute and were adaptively fed for one week. The health status of the experimental fish and the water quality parameters were monitored daily, confirming that no abnormal symptoms were observed during clinical observations.

### 2.3. Tested compounds

KMPS, THPS, BS, and CS were purchased from Henan Nanhua Qianmu Biotechnology Co., Ltd. (Q/SDSA1100-2021, Zhengzhou, China), Nanjing Sailte Biotechnology Co., Ltd. (Q/320116SRT08-2017, Nanjing, China), Henan Nanhua Qianmu Biotechnology Co., Ltd. (Q/HJA0001-2018, Zhengzhou, China), Bu Rui Biotechnology Co., Ltd. (Q/PRSW02-2020), respectively.

### 2.4. Experimental diets

The experimental diets included five main raw materials of basic feed, such as fish meal, soybean meal, rapeseed meal, cellulose, and flour. **Table 1** lists the raw material composition and nutrition level of these experimental diets. The study determined the crude protein, crude fat, and crude fiber of the experimental diets employing the Kjeldahl method (GB/T5009.5, China), Soxhlet extraction method (GB/T5009.6, China), and acid-base digestion method (GB/T5515, China), respectively. The other nutrition level values were calculated.

### 2.5. Feeding experiment

After one week of adaptive feeding, 1500 GIFT tilapia were randomly divided into five groups (3 replicates/group,100 tilapias/replicate; n = 100 × 3 = 300), including blank control, KMPS, THPS, BS, and CS groups. Each group was individually reared in a 5 $m^3$ cement pond with fresh water at 25±3.0˚C. With the exception of the control group, the experimental cement ponds were treated with an aqueous solution containing KMPS, THPS, BS, and CS. This treatment was administered once every seven days, resulting in concentrations of 1.5, 1, 5, and 2 mg/L respectively. These concentrations were chosen based on the recommended clinical dosage provided in the instructions for each preparation. The water in the experimental cement ponds remained unchanged throughout the experiment. It was refilled as necessary to compensate for water loss due to evaporation, ensuring that the water volume remained constant following the initial water level markings.

**Table 1. Raw material composition and nutrition level of experimental diet.**

| Raw material composition (g/kg dry weight) | | Nutrition level (%) | |
|---|---|---|---|
| Fish meal | 220.00 | Crude protein | 43.10 |
| Soybean meal | 160.00 | Crude fiber | 4.50 |
| Rapeseed meal | 120.00 | Crude ash content | 15.00 |
| Cellulose | 160.00 | Crude fat | 5.06 |
| Flour | 270.00 | Calcium | 3.05 |
| Shrimp shell powder | 10.00 | Total phosphorus | 1.52 |
| Squid cream | 20.00 | Lysine | 2.56 |
| Soybean phospholipid oil | 20.00 | | |
| Choline chloride | 5.00 | | |
| Salt | 5.00 | | |
| Vitamin premix[1] | 5.00 | | |
| Mineral premix[2] | 5.00 | | |

Vitamin premix: riboflavin 45.00 mg; thiamine 25.00 mg; Vit K 10.00 mg; inositol 200.00 mg; pyridoxine hydrochloride 10.00 mg; Vit B12 2.00 mg; calcium pantothenate 60.00 mg; biotin 1.30 mg; Vit A 820000.00 IU; Vit D 50000.00 IU; nicotinic acid 200.00 mg; folic acid 20.00 mg; Vit E 12.00 mg; Vit C 16.00 mg.

Per gram mineral premix: zinc sulfate 60.00 mg; sodium fluoride 50.00 mg; cobalt chloride 50.00 mg; potassium chloride 70.00 mg; 20.00 mg; calcium dihydrogen phosphate 80.00 mg; sulfate 80.00 mg; manganese sulfate 30.00 mg; ferrous sulfate 80.00 mg; calcium chloride 190.00 mg; copper sulfate 50.00 mg.

Tilapia were provided with a standard diet comprising 2–3% of their body weight, which was subsequently adjusted based on their actual feed intake. This feeding regimen was administered twice daily, at 10:00 and 18:00, for 35 days. Following 1 h feeding, the remaining unconsumed feed was collected, subjected to a drying process, and subsequently weighed to determine the precise daily feed intake for each respective experimental group. The daily monitoring and recording of the activity levels and dietary consumption of GIFT tilapia were conducted in each experimental group.

## 2.6. Water quality parameter evaluation

The study used the detection instrument (HANNA Company of Italy) to continuously measure the water temperature, dissolved oxygen (DO), pH, ammonia nitrogen, active phosphate, nitrite, total organic carbon (TOC), and chemical oxygen demand (COD) daily from 11:00 to 15:00 throughout the experiment.

## 2.7. Growth performance

The growth performance parameters were evaluated using weight gain rate (WGR [1]), specific growth rate (SGR[2]), condition factor (CF [3]), feed conversion ratio (FCR[4]), spleen index (SI [5]), hepato-somatic index (HSI[6]). The calculation formula utilized was as follows.

$$WGR = (FBW-IBW)/IBW \times 100 \qquad (1)$$

$$SGR = (lnFBW-lnIBW)/T \times 100 \qquad (2)$$

$$CF = FBW/FBL^3 \times 100 \qquad (3)$$

$$FCR = FW/(FBW-IBW) \qquad (4)$$

$$SI = SW(g)/FBW(g) \times 100 \qquad (5)$$

$$HSI = LW(g)/FBW(g) \times 100 \qquad (6)$$

Where IBW and FBW (g) refer to the initial and final body weight of GIFT tilapia, respectively. T (day, d) refers to the experimental feeding time. FBL(cm) refers to the final body length of GIFT tilapia. FW (g) refers to the feed weight of each group of tilapia. SW and LW (g) refer to spleen and liver weights, respectively.

## 2.8. Immunological parameters

Blood samples were obtained from the caudal vein of GIFT tilapia on two specific days, the 18th and 35th day following administration. The blood collection was performed below the lateral line of the anal fin of GIFT tilapia. A volume of approximately 3–4 mL of blood was extracted from each fish and subsequently divided into two equal portions. A portion of blood was sub-packed into a tube containing dipotassium ethylenediamine tetraacetate (K2-EDTA) to test the blood routine index within a time frame of 4 h. The remaining portion of the blood sample was transferred into a sterile 2 mL aseptic Eppendorf tube and positioned obliquely within a test tube overnight. Subsequently, the tube was centrifuged at 4˚C, with an rcf of 956, for 10 min. Subsequently, serum was collected and preserved at -80˚C and was then used to determine aspartate aminotransferase (AST), alanine transaminase (ALT), creatinine (CRE), blood urea nitrogen (BUN), total protein (TP), albumin (ALB), acid phosphatase (ACP), alkaline phosphatase (AKP), superoxide dismutase (SOD), lysozyme (LZM), nitric oxide (NO) and total antioxidant capacity (T-AOC). These kits employed in this study to assess these parameters was procured from Nanjing Jiancheng Bioengineering Institute (Nanjing, China) and was tested according to the method provided by the kit.

## 2.9. Cytokines genes expression

Four liver or spleen tissue samples were taken from each replicate/group (n = 12 tilapias/group). The samples were ground and preserved with RNA Keeper tissue stabilizer (Vazyme, China) to determine cytokine. The study employed the Trizol reagent to extract the total RNA of liver or spleen tissues, followed by detecting the RNA purity utilizing 1.5% agarose gel electrophoresis and then determining the total RNA concentration of the extracted samples. Subsequently, the RNA was reversely transcribed into cDNA fragments following the AMB reverse transcription kit instructions. Additionally, the mRNA expression levels of immune-related cytokine of GIFT tilapia were detected strictly per the instructions of the Blas-Taq2XqPCRMasterMix kit (Merck, China). **Table 2** shows the cytokine primer sequences that were synthesized by Shanghai Shenggong Bioengineering Service Co., Ltd. (Shanghai, China).

## 2.10. Histopathology

On the 35th day of the feeding experiment, liver and spleen tissue samples were randomly collected from GIFT tilapia from each replicate per group (12 tilapias/group). The collected tissue samples were then fixed using a 10% formaldehyde solution for 48 h. Subsequently, the samples were subjected to dehydration, transparency, embedding, slicing, and staining with hematoxylin and eosin (H&E) using established histological techniques [18]. Histomorphological changes were observed through an optical microscope.

**Table 2. Primer sequences for Q-PCR.**

| Gene | Primer sequences | Accession Number |
|---|---|---|
| TNF-α | F: ATGTGCCGTGCTGTCGCT | XM_003456260.4 |
| | R: GCTATGGGAAACAGGAAAGAAGTG | |
| IL-1β | F: TTCACCAGCAGGGATGAGATT | KF747686.1 |
| | R: TGGAGGGTTGGCTTGTCG | |
| Interferon-gamma (IFN-γ) | F: GATCTTCATGGGTGGTGTTTTG | XM_003448130.1 |
| | R: GGTAGCGAGCCTGAGTTGTTG | |
| Beta-actin (β-actin) | F: ACCTGAGCGTAAATACTCCGTCT | EF026001.1 |
| | R: AAGTTGTTGGGCGTTTGGTT | |

### 2.11. Data analysis

Statistical comparisons of experimental data were performed by one-way analysis of variance (ANOVA) using SPSS 22.0 software (IBM, USA). Duncan's Multiple Range test was used to identify significant differences. Data are presented as mean ± standard error. Lowercase letters (a, b, c, d, and e) denote significant differences between different sampling groups (determined by Duncan's test, $P < 0.05$).

## 3. Results

### 3.1. Water quality parameter evaluation

Fig 1 shows the effects of the four water quality regulators on the growth water quality parameters of tilapia. The results showed that KMPS, THPS, BS, and CS could reduce the water contents of ammonia nitrogen, active phosphate, nitrite, TOC, and COD to different degrees, particularly BS, which had the most significant effect (Fig 1C–1G). Meanwhile, BS and CS significantly maintained the stability of DO and pH of the water environment (Fig 1A and 1B). Although KMPS and THPS had not significantly maintained the stability of water DO, KMPS could effectively stabilize water pH (Fig 1B).

### 3.2. Growth performance

During the experiment, tilapia had no obvious pathological changes and abnormal death, and its activity, food intake and body color were normal. Table 3 demonstrates the effects of the four water quality regulators on the growth performance of GIFT tilapia. On day 18, the four water quality regulators groups exhibited significantly increased FBW, WGR, CF, and SGR of GIFT tilapia ($P < 0.05$) compared to the control group. Meanwhile, the THPS, BS, and CS groups exhibited significantly decreased FCR ($P < 0.05$). Additionally, the KMPS and THPS groups exhibited significantly decreased HSI ($P < 0.05$). On day 35, the FBW of tilapia in the four water quality regulator groups increased significantly ($P < 0.05$). At the same time, the BS and CS groups showed significantly increased WGR, SGR, SI, and HSI of tilapia ($P < 0.05$) while showing significantly decreased FCR ($P < 0.05$). The KMPS and THPS groups showed a significant decrease in the HSI of tilapia ($P < 0.05$).

### 3.3. Blood biochemical

Table 4 shows the effects of the four water quality regulators on the blood biochemistry of GIFT tilapia. On day 18, ALB and TP in the four water quality regulators groups, as well as the CRE, ALT, and AST in KMPS and THPS groups, were significantly increased ($P < 0.05$) compared to the control group.

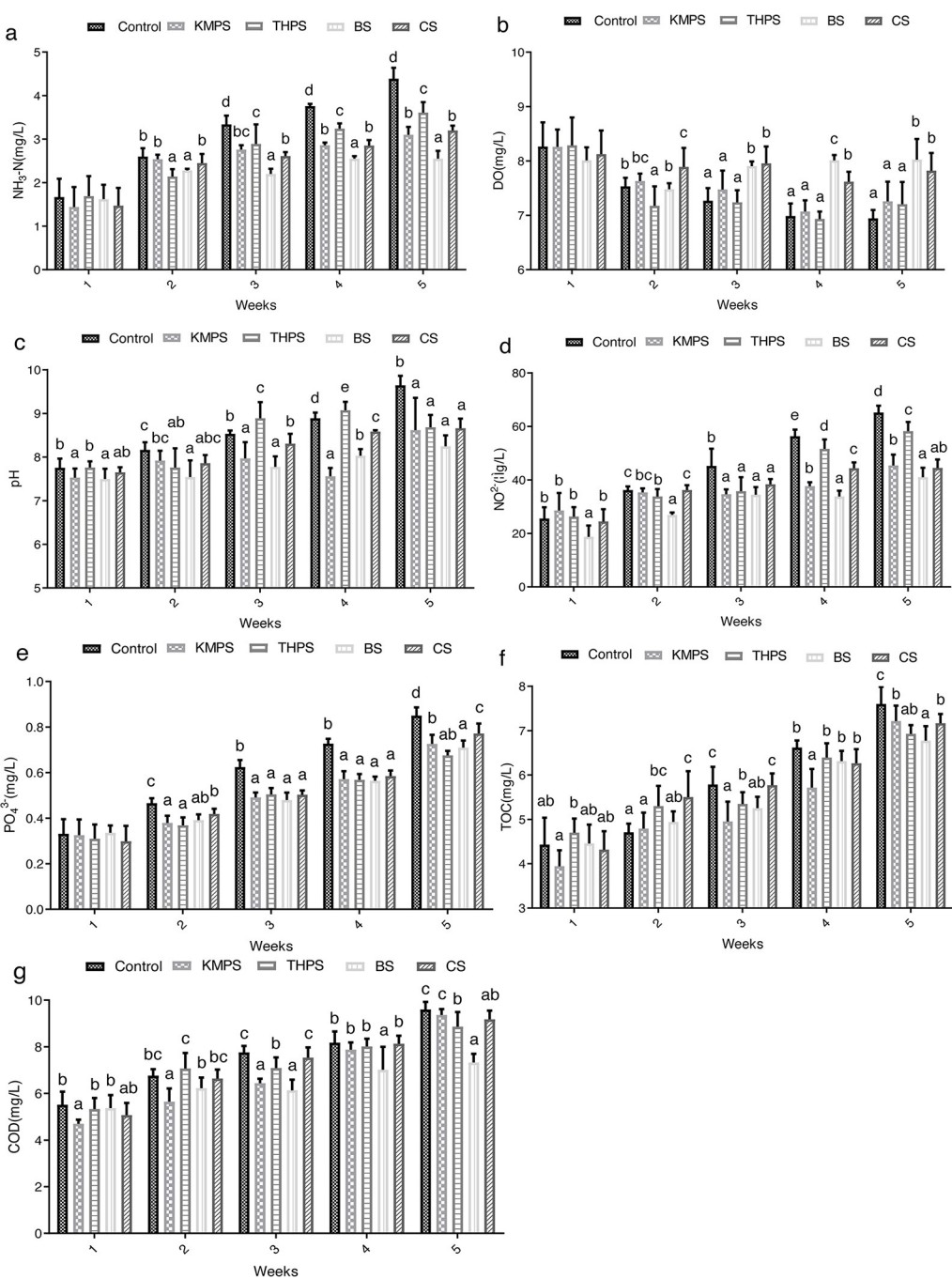

**Fig 1. Water quality parameters during the trial.** Values are presented as means ± SD (n = 7). The letters a, b, c, d, e, f, and g refer to DO: dissolved oxygen, pH, $NO_2^-$: nitrite, COD: chemical oxygen demand, $NH_3$-N: ammonia nitrogen, TOC: total organic carbon, and $PO_4^{3-}$: active phosphate, respectively. Bars with different lowercase letters are statistically different (one-way ANOVA, $P < 0.05$, and subsequent post hoc multiple comparisons with the SNK test).

On day 35, the CRE, BUN, AST, and ALT in KMPS and THPS groups were significantly higher than the control group ($P < 0.05$). Moreover, ALB and TP in BS and CS groups were significantly increased ($P < 0.05$).

**Table 3. Effects of the four water quality regulators on growth performance and organ index of GIFT tilapia.**

| Times | Test item | Control | KMPS | THPS | BS | CS |
|---|---|---|---|---|---|---|
| 18d | $W_0$ (g) | 479.41±45.45 | 475.50±53.81 | 479.55±48.65 | 478.56±51.42 | 478.25±51.95 |
| | $W_{18}$ (g) | 577.18±26.07[a] | 609.12±48.03[b] | 597.43±49.86[b] | 616.50±51.83[b] | 613.00±53.62[b] |
| | WGR (%) | 20.70±1.45[a] | 27.06±2.02[b] | 25.64±2.49[b] | 28.56±2.81[b] | 28.09±2.21[b] |
| | SGR (%/d) | 1.05±0.15[a] | 1.33±0.13[b] | 1.27±0.15[b] | 1.38±0.18[b] | 1.36±0.19[b] |
| | CF (g/cm$^3$) | 3.04±0.13[a] | 3.66±0.30[c] | 3.38±0.24[b] | 3.32±0.26[b] | 3.37±0.31[b] |
| | FCR | 1.28±0.11[a] | 1.02±0.11[a] | 1.11±0.10[a] | 1.05±0.05[a] | 1.06±0.01[a] |
| | SI (%) | 0.12±0.01[a] | 0.11±0.01[a] | 0.11±0.01[a] | 0.12±0.01[a] | 0.12±0.01[a] |
| | HSI (%) | 1.35±0.16[b] | 1.12±0.09[a] | 1.23±0.17[a] | 1.36±0.21[b] | 1.31±0.16[b] |
| 35d | $W_{35}$ (g) | 661.53±32.52[a] | 701.90±38.22[bc] | 691.90±48.27[b] | 749.35±53.35[d] | 728.85±45.86[cd] |
| | WGR (%) | 13.91±1.60[a] | 14.50±1.24[ab] | 14.63±1.00[ab] | 21.45±1.65[c] | 18.90±1.48[bc] |
| | SGR (%/d) | 0.73±0.15[a] | 0.74±0.11[a] | 0.74±0.18[a] | 1.07±0.10[b] | 0.95±0.14[ab] |
| | CF (g/cm$^3$) | 3.56±0.23[a] | 3.57±0.13[a] | 3.55±0.14[a] | 3.66±0.17[a] | 3.60±0.21[a] |
| | FCR | 1.49±0.16[b] | 1.66±0.16[b] | 1.83±0.05[b] | 1.10±0.13[a] | 1.20±0.17[a] |
| | SI (%) | 0.11±0.01[a] | 0.11±0.02[a] | 0.11±0.01[a] | 0.15±0.01[b] | 0.13±0.01[b] |
| | HSI (%) | 1.33±0.06[b] | 1.15±0.11[a] | 1.23±0.17[a] | 1.59±0.08[c] | 1.64±0.05[c] |

The values are presented as mean ± standard errors (n = 40). Values with different superscript letters indicate significant differences ($P < 0.05$) among all the treatments.

## 3.4. Blood physiology

**Table 5** demonstrates the effects of the four water quality regulators on the blood physiology of GIFT tilapia. After continuous administration for 18 days, the total blood platelet count of tilapia in the KMPS and THPS groups was significantly higher than in the control group ($P < 0.05$).

On day 35, the total number of leukocytes, lymphocytes, intermediate cells, neutrophils, and hemoglobin of GIFT tilapia in the BS and CS groups was significantly higher than in the control group ($P < 0.05$). The total number of leukocytes, lymphocytes, intermediate cells, and neutrophils of GIFT tilapia in KMPS and THPS was significantly higher than in the control,

**Table 4. Effects of the four water quality regulators on blood biochemical indexes of GIFT tilapia.**

| Times | Test item | Control | KMPS | THPS | BS | CS |
|---|---|---|---|---|---|---|
| 18d | CRE (mmol/L) | 19.50±3.07[a] | 29.74±4.77[c] | 24.56±3.51[b] | 19.41±3.85[a] | 17.00±4.63[a] |
| | BUN (mmol/L) | 1.65±0.06 | 1.70±0.12 | 1.68±0.14 | 1.68±0.06 | 1.65±0.12 |
| | ALB (g/L) | 13.97±2.03[a] | 18.01±1.36[b] | 17.42±1.88[b] | 17.24±1.02[b] | 20.30±1.21[c] |
| | ALT (IU/L) | 35.77±2.81[a] | 41.26±3.02[b] | 42.09±4.05[b] | 34.14±2.38[a] | 36.57±2.33[a] |
| | AST (IU/L) | 19.24±3.56[a] | 26.23±2.08[b] | 24.17±3.43[b] | 20.06±3.03[a] | 19.78±3.30[a] |
| | TP (g/L) | 44.22±0.78[a] | 47.95±2.65[b] | 47.11±2.29[b] | 46.72±2.35[b] | 47.09±2.00[b] |
| 35d | CRE (mmol/L) | 25.96±2.18[a] | 38.08±2.24[b] | 34.65±2.83[b] | 30.23±2.01[ab] | 25.42±2.45[a] |
| | BUN (mmol/L) | 1.43±0.16[a] | 2.56±0.32[b] | 2.68±0.22[b] | 1.51±0.16[a] | 1.27±0.24[a] |
| | ALB (g/L) | 13.49±1.19[a] | 12.50±1.06[a] | 14.16±1.55[a] | 18.16±1.13[b] | 17.18±1.45[b] |
| | ALT (IU/L) | 35.30±1.79[a] | 44.89±2.20[b] | 46.92±3.17[b] | 39.47±2.89[ab] | 40.82±2.46[ab] |
| | AST (IU/L) | 18.82±2.30[a] | 22.98±3.02[b] | 23.12±2.81[b] | 20.19±3.09[a] | 19.78±1.95[a] |
| | TP (g/L) | 47.21±2.14[a] | 47.94±1.06[a] | 48.41±0.61[a] | 50.51±1.20[b] | 50.24±0.93[b] |

The values are presented as mean ± standard errors (n = 24). Values with different superscript letters indicate significant differences ($P < 0.05$) among all the treatments.

**Table 5. Effects of the four different water quality regulators on blood physiology of GIFT tilapia.**

| Times | Test item | Control | KMPS | THPS | BS | CS |
|---|---|---|---|---|---|---|
| 18d | Total leukocyte count (10^9/L) | 257.49±19.92 | 249.11±12.13 | 269.22±15.85 | 257.15±23.98 | 263.30±16.88 |
| | Lymphocyte (10^9/L) | 227.28±14.27 | 215.10±9.19 | 232.06±14.24 | 224.62±22.24 | 225.85±11.46 |
| | Intermediate cell (10^9/L) | 8.57±1.58 | 14.94±17.29 | 9.79±1.34 | 8.96±1.23 | 10.15±1.46 |
| | Neutrophils (10^9/L) | 21.65±5.59 | 25.82±8.10 | 27.38±5.30 | 23.58±4.74 | 27.31±4.77 |
| | Lymphocyte ratio (%) | 88.36±2.00 | 86.43±3.14 | 86.21±2.12 | 87.32±2.14 | 85.83±1.53 |
| | Intermediate cell ratio (%) | 3.30±0.37 | 3.67±0.56 | 3.64±0.36 | 3.48±0.36 | 3.84±0.31 |
| | Neutrophil ratio (%) | 8.34±1.66 | 9.90±2.59 | 10.15±1.79 | 9.20±1.79 | 10.33±1.22 |
| | Hemoglobin content (g/L) | 79.5±15.53 | 78.40±4.67 | 78.50±12.47 | 85.90±9.61 | 83.80±17.45 |
| | Mean red blood cell volume (fL) | 56.17±5.04 | 58.50±6.00 | 54.99±4.06 | 57.52±5.17 | 59.45±5.43 |
| | Total platelet count (10^9/L) | 618.00±99.52[a] | 732.10±67.27[b] | 725.10±86.94[b] | 701.00±109.66[ab] | 660.70±100.44[ab] |
| 35d | Total leukocyte count (10^9/L) | 357.87±17.88[a] | 489.87±19.22[c] | 469.35±42.84[c] | 397.80±35.58[b] | 410.73±46.28[b] |
| | Lymphocyte (10^9/L) | 311.18±11.46[a] | 417.18±13.42[c] | 399.90±29.69[c] | 353.96±27.29[b] | 364.44±32.39[b] |
| | Intermediate cell (10^9/L) | 9.43±1.21[a] | 19.83±1.89[c] | 19.47±3.12[c] | 15.02±2.10[b] | 16.56±4.63[b] |
| | Neutrophils (10^9/L) | 20.86±3.55[a] | 45.86±3.75[c] | 47.98±6.84[c] | 27.80±3.74[b] | 29.55±2.71[b] |
| | Lymphocyte ratio (%) | 87.41±2.94 | 86.65±4.11 | 85.46±4.59 | 89.97±4.20 | 87.80.00±2.54 |
| | Intermediate cell ratio (%) | 2.87±0.43 | 2.87±0.41 | 3.01±0.42 | 3.14±0.25 | 3.55±0.60 |
| | Neutrophil ratio (%) | 8.17±0.58 | 9.28±0.70 | 9.88±0.70 | 6.89±1.11 | 7.65±1.01 |
| | Hemoglobin content (g/L) | 89.40±6.16[a] | 87.90±8.54[a] | 104.20±12.38[bc] | 108.00±8.70[c] | 99.60±5.23[b] |
| | Mean red blood cell volume (fL) | 54.70±6.86 | 54.70±6.86 | 58.65±5.54 | 57.76±7.56 | 59.41±6.62 |
| | Total platelet count (10^9/L) | 66.80±5.79 | 67.60±5.22 | 68.00±7.42 | 70.90±4.25 | 73.60±7.30 |

The values are presented as mean ± standard errors (n = 15). Values with different superscript letters indicate significant differences ($P < 0.05$) among all the treatments.

BS, or CS groups ($P < 0.05$). In addition, THPS also significantly increased the hemoglobin content of GIFT tilapia ($P < 0.05$).

### 3.5. Immune function and antioxidant capacity

Table 6 indicates the effects of the four water quality regulators on the immune function and antioxidant capacity of GIFT tilapia. After continuous administration for 18 days, AKP, SOD,

**Table 6. Effects of the four water quality regulators on serum immune function of GIFT tilapia.**

| Times | Test item | Control | KMPS | THPS | BS | CS |
|---|---|---|---|---|---|---|
| 18d | ACP (U/100 mL) | 9.79±0.98[a] | 10.04±0.80[a] | 10.47±0.96[a] | 9.24±1.03[a] | 9.89±1.66[a] |
| | AKP (U/100 mL) | 6.79±0.72[a] | 9.73±1.06[b] | 11.28±1.09[b] | 9.47±1.41[b] | 9.49±1.47[b] |
| | SOD (U/mL) | 37.16±3.08[a] | 55.97±2.85[b] | 54.15±1.98[b] | 57.27±3.57[b] | 54.42±3.21[b] |
| | T-AOC (mM) | 0.22±0.05[a] | 0.37±0.04[b] | 0.40±0.06[bc] | 0.49±0.06[c] | 0.34±0.04[b] |
| | NO (μmol/L) | 1.15±0.13 | 1.23±0.17 | 1.19±0.14 | 1.31±0.10 | 1.33±0.15 |
| | LZM (U/mL) | 177.30±4.86 | 183.41±8.70 | 177.52±9.20 | 184.45±6.70 | 186.04±14.81 |
| 35d | ACP (U/100 mL) | 9.00±1.62[a] | 9.17±1.22[a] | 9.32±1.49[a] | 10.40±1.16[b] | 10.44±0.61[b] |
| | AKP (U/100 mL) | 7.52±1.57[a] | 7.65±1.59[a] | 8.77±1.81[a] | 9.58±1.47[b] | 9.20±1.61[b] |
| | SOD (U/mL) | 35.01±2.80[a] | 46.15±3.85[bc] | 50.18±4.12[c] | 43.23±3.87[b] | 46.59±3.43[bc] |
| | T-AOC (mM) | 0.64±0.08[b] | 0.48±0.07[a] | 0.53±0.14[ab] | 0.78±0.12[c] | 0.80±0.08[c] |
| | NO (μmol/L) | 1.72±0.19[a] | 2.85±0.21[c] | 2.62±0.20[c] | 1.96±0.14[b] | 1.97±0.24[b] |
| | LZM (U/mL) | 196.25±13.82[a] | 189.09±8.22[a] | 204.22±6.37[a] | 227.20±6.91[c] | 214.30±6.36[b] |

The values are presented as mean ± standard errors (n = 24). Values with different superscript letters indicate significant differences ($P < 0.05$) among all the treatments.

and T-AOC levels in the four different water quality regulator groups were significantly higher than in the control ($P < 0.05$). Additionally, the T-AOC level in the BS group was significantly higher than in the KMPS or CS group ($P < 0.05$). The effects of each experimental group on LZM and NO showed no significant difference ($P > 0.05$).

On day 35, BS and CS significantly increased ACP, AKP, NO, LZM, SOD, and T-AOC levels ($P < 0.05$) compared to the control group. Moreover, the KMPS and THPS significantly increased SOD and NO levels ($P < 0.05$) while significantly decreasing T-AOC serum levels, with KMPS exhibiting the most significant difference ($P < 0.05$).

### 3.6. Cytokine genes expression

**Figs 2 and 3** show the mRNA expression levels of TNF-α, IL-1β, and IFN-γ of the liver or spleen are shown in. On day 18, the TNF-α, IL-1β, and IFN-γ mRNA expression levels in the BS and CS groups were significantly higher than in the control group ($P < 0.05$). Additionally, the TNF-α in the BS and CS groups was significantly higher than in the control group ($P < 0.05$).

On day 35, The mRNA expression levels of TNF-α, IL-1β, and IFN-γ of the liver or spleen in the BS and CS groups exhibited significantly higher levels than in the control group ($P < 0.05$). Furthermore, the TNF-α and IL-1β levels of the liver in the KMPS and THPS groups were significantly higher than in all other experimental groups ($P < 0.05$). Similarly, the TNF-α and IL-1β levels of the spleen in the KMPS group were significantly higher than in all other experimental groups ($P < 0.05$). The TNF-α level of the spleen in the THPS group was significantly higher than in the control group ($P < 0.05$).

### 3.7. Histopathological observation

No significant histopathological alterations were observed in the liver or spleen tissue of the control, BS, and CS groups on day 35. The tissue structure of the liver or spleen was clear, and

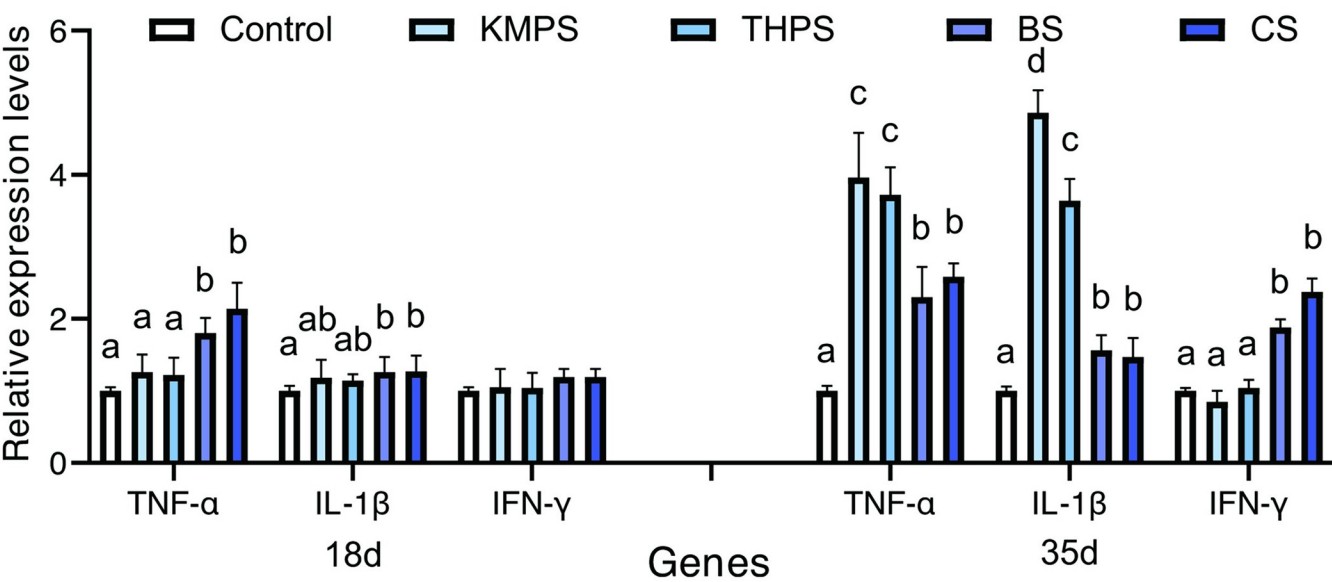

**Fig 2. Effects of the four different water quality regulators on the expression of immune-related factors in tilapia liver.** The values are presented as mean ± standard errors (n = 4). Values with different superscript letters indicate significant differences ($P < 0.05$) among all the treatments.

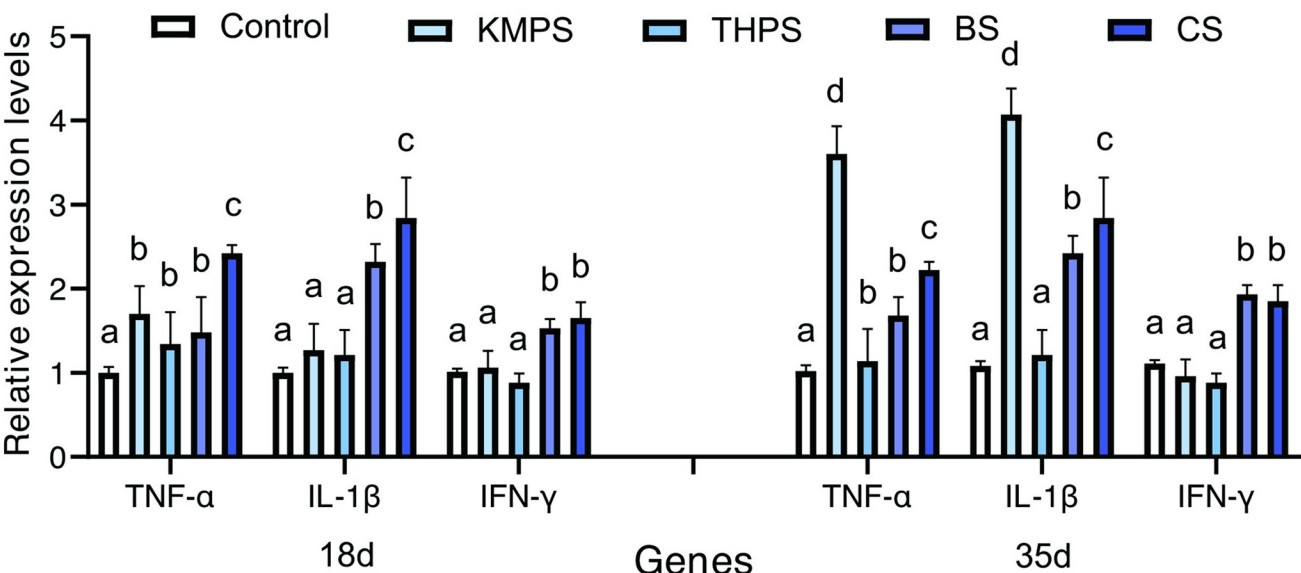

**Fig 3. Effects of the four different water quality regulators on the expression of immune-related factors in tilapia spleen.** The values are presented as mean ± standard errors (n = 4). Values with different superscript letters indicate significant differences ($P < 0.05$) among all the treatments.

the cells were evenly distributed, arranged orderly, intact, and with full nuclei (**Figs 4A, 4D, 4E, 5A, 5D and 5E**). Nevertheless, the liver of tilapia in KMPS and THPS groups showed significant pathological changes, such as tissue vacuolation to varying degrees, cell shrinkage, cell nucleus shift, and nucleolar pyknosis (**Fig 4B and 4C**). Nevertheless, the spleen showed no significant histopathological changes (**Fig 5B and 5C**).

### 3.8. Relativity analysis

The Pearson correlation between water environment factors and immune-related factors of tilapia was analyzed by the omicshare cloud tool (https://www.omicshare.com/tools/). The results showed that the levels of COD, TOC, active phosphate, pH, nitrite, and ammonia nitrogen in GIFT tilapia growing water environment were positively correlated to T-AOC in serum while negatively correlated to ALB and SOD (***P* < 0.05; Fig 6**). Active phosphate and ammonia nitrogen were significantly negatively correlated to AKP and ACP ($P < 0.05$). A significant positive correlation was found between DO and ALB, AKP, LZM, and IFN-γ in the liver ($P < 0.05$).

Additionally, we analyzed the correlation between environmental factors and blood biochemistry (**Fig 6**). The results showed that even though the blood biochemistry of tilapia changed greatly during the experiment, the correlation between environmental factors and the contents of CRE, BUN, AST and ALT in tilapia blood was not significant ($P > 0.05$).

### 4. Discussion

The growth rate and immune level of tilapia in aquaculture are significantly influenced by the quality of water. Previous research has indicated that ammonia, nitrogen, and nitrite are significant toxic substances that arise from the feces and residual feed of aquatic animals during

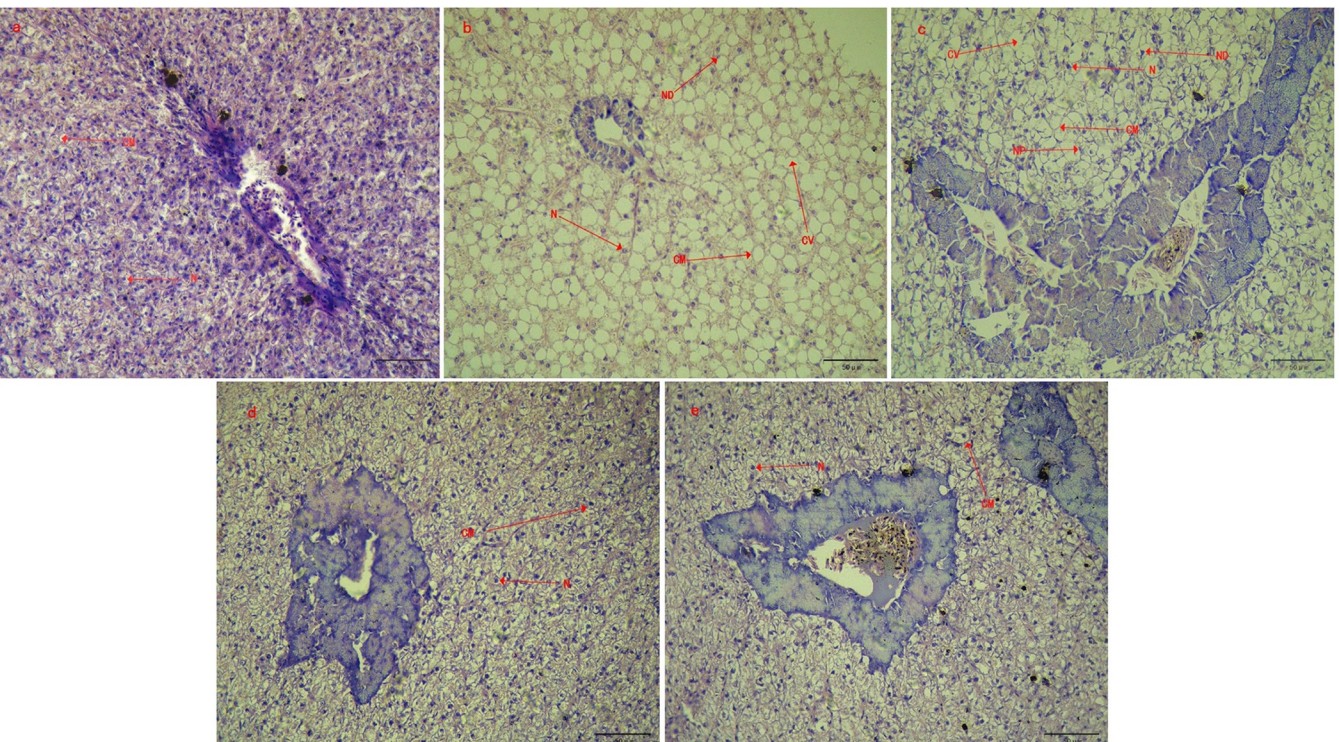

**Fig 4. Pathomorphological observation of tilapia liver tissue.** a: Control, b: KMPS, c: THPS, d: BS and e: CS. The N: nucleus; CM: cell membrane; CV: cell vacuolization; ND: nuclear deviation; NP: nuclear pyknosis; stained with hematoxylin and eosin (H&E), 400×; scale bars, 50 μm.

aquaculture. These substances can potentially result in the mortality of fish and shrimp, particularly when their concentrations exceed certain thresholds [19, 20]. As a result, reducing these deleterious substances has emerged as the central objective and challenge in water quality regulation. In addition, the content of organic matter, biological quantity, DO, pH, and COD in water hold great significance to tilapia growth [21]. Briefly, DO has important effects on tilapia growth performance, feed efficiency, liver, and immunity [12]. The deposition of ammonia nitrogen, total nitrogen, total phosphorus, TOC, nitrite, and nitrate will expedite the process of eutrophication and have implications for the economic viability of tilapia aquaculture [21, 22]. Currently, KMPS, THPS, BS, and CS were found to potentially promote the degradation of underwater residues and harmful substances to some extent. Particularly, both KMPS and THPS can purify water mainly by inhibiting or killing some harmful bacteria and inhibiting algae growth [5]. Differently, research shows that BS mainly promotes the decomposition of harmful substances in sediment or improves the microflora of the water body by producing various enzymes, improving water quality [20]. Moreover, CS can absorb many harmful substances as a good biodegradability, which is significant in promoting the decomposition of harmful substances in water [14]. Herein, KMPS, THPS, BS, and CS could reduce the increasing rate of ammonia nitrogen, PH, nitrite, active phosphate, TOC, and COD, besides maintaining DO content in GIFT tilapia growing environment, with BS or CS exhibiting the most significant effect. However, the regulatory effect of KMPS and THPS on water quality is not significant, which is related to their killing effect on microflora in the water environment [7]. The results showed that the frequent usage of KMPS and THPS leads to the destruction of the microbial community and the imbalance of steady state in the water environment [7, 23]. Growth performance is the most intuitive manifestation of the water quality effect on tilapia

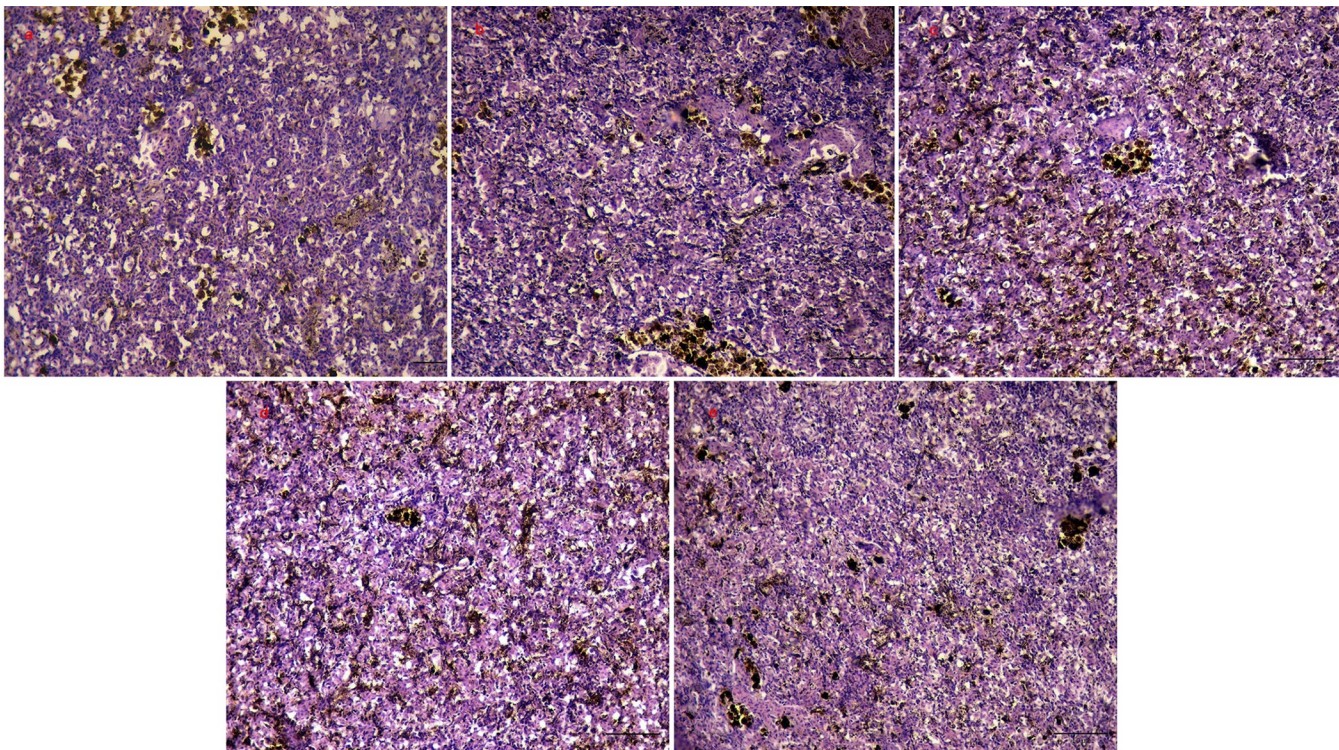

**Fig 5. Pathomorphological observation of tilapia spleen tissue.** a: Control, b: KMPS, c: THPS, d: BS, and e: CS. The spleen was stained with hematoxylin and eosin (H&E), 400×; scale bars, 50 μm.

growth [24]. Herein, BS and CS can significantly improve the growth performance of GIFT tilapia, reduce CF, and contribute significantly to growth enhancement. According to the findings of Liu *et al.*, *bacillus subtilis* can provide a better intestinal flora environment for tilapia as well as improve intestinal absorption and metabolism of nutrients to promote tilapia growth

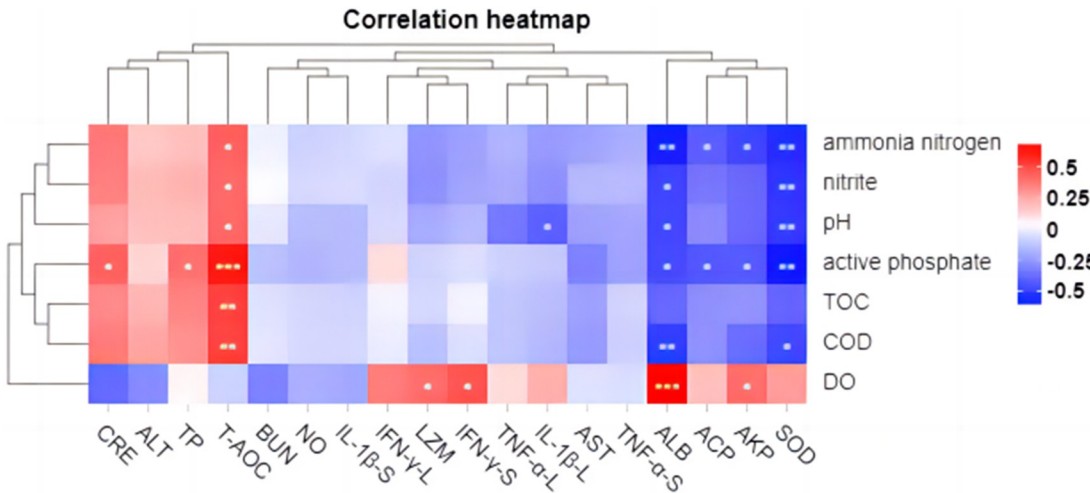

**Fig 6. Correlation analysis between environmental factors and immune factors.** Red squares indicate positive correlations, whereas blue squares indicate negative correlations. Asterisks within the different squares indicate significance, * $P < 0.05$, ** $P < 0.01$. The suffix L indicates the index in the liver, and the suffix S indicates the index in the spleen.

[25]. Similarly, CS can promote tilapia growth by increasing feed conversion rate [26, 27]. Currently, there is a lack of reported literature on applying KMPS and THPS in tilapia culture. However, the strong killing effect of these two regulators on pathogens plays an important role in the degradation of bait and harmful substances in pond sludge and feces [5, 7]. The experimental results show that using KMPS and THPS over a brief duration of 18 days yields a significant enhancement in growth. However, as the frequency of usage increases, the efficacy of KMPS and THPS diminishes, potentially due to their lethal impact. In our experiment, the CRE and BUN serum content of KMPS and THPS-treated GIFT tilapia increased significantly, which may be resulted from the metabolic disorders caused by renal tissue damage [28]. The results showed that KMPS and THPS had certain effects on the renal function of GIFT tilapia. In our experiment, KMPS and THPS increased ALT and AST serum contents of GIFT tilapia, suggesting that these two water quality regulators have certain side effects on the liver and kidneys of GIFT tilapia [29]. Moreover, ALB and TP are the most intuitive biochemical indexes that can be used to measure nonspecific immunity and liver function [30]; their contents in serum are positively correlated with protein metabolism and immune function [31]. The results showed that BS, CS, KMPS, and THPS could increase TP and ALB serum contents in GIFT tilapia, which could be gradually weakened due to the frequent use of KMPS and THPS. The production of immunoglobulin and ALB is closely related to liver and spleen [30], which are considered important immune organs of fish. The organ index is one of the important indicators for measuring organ perfection, and its size can reflect the immune ability of the body. Typically, the higher the organ index, the stronger the immunity of the body. In addition, the size of organ index is closely related to the growth and metabolism level of tilapia [2]. Similar to this study, Li *et al* found that reducing fatty acid β-oxidation can effectively improve glucose catabolism and liver health of juvenile tilapia fed with high-starch diets, which can promote metabolism and enhance immunity [30]. Here, the SI and HSI of GIFT tilapia in BS and CS groups were significantly increased, indicating that BS and CS could significantly improve the SI and HSI to improve the metabolic function and immune function of GIFT tilapia.

The innate immune system of tilapia includes cellular and humoral immune responses, which work together to provide protection against microbial infections [32]. The cellular response involves multiple immune functions, but it is inevitable that they all depend on the number and proportion of immune cells [33]. Immune cells exhibit a close association with immune response, growth, and metabolism, while also serving as a partial indicator of immune function potency [34]. White blood cells, lymphocytes, neutrophils, and intermediate cells are important immune cells in the fish body, which participate in various immune responses in the fish body [35]. Therefore, the content and activity of immune cells in the blood can well reflect the immune levels of the body [36]. As the main component of red blood cells, hemoglobin is mainly involved in the transport of oxygen and carbon dioxide in the body, which can directly reflect the ability and metabolism of the body to transport oxygen [37]. In our experiment, BS and CS can effectively increase the number of leukocytes, lymphocytes, neutrophils, intermediate cells, platelets, and hemoglobin of GIFT tilapia, besides up-regulating the level of the cellular immune response. The difference is that the total number of leukocytes, lymphocytes, intermediate cells and neutrophils in the blood of tilapia in KMPS and THPS groups is abnormally higher than the normal value, suggesting that there may be inflammatory damage in the body. Humoral responses involve several unspecific enzymes or factors, such as lysozyme, superoxide, and dismutase, among others. These components function to eradicate pathogens either through direct pathogen killing or by impeding pathogen growth and dissemination [38]. AKP and ACP are important immune indexes that participate in several metabolic and immune activities in the body [39]. AKP is a key enzyme in metabolism and antioxidation, which participates in the regulation of phosphate groups in the body

[40]. Similarly, ACP is an important enzyme in material metabolism and signal transduction and plays an important role in the metabolism of phosphate groups, nucleic acids, proteins, and lipids [4]. T-AOC represents a highly intuitive embodiment of the antioxidant capacity of the body. In conjunction with SOD, T-AOC actively participates in the process of scavenging free radicals within the body [41]. LZM can kill the pathogen by destroying the cell wall through a series of reactions [42]. In addition, LZM can directly bind to negatively charged virus proteins and form double salt with nucleic acid substances, resulting in virus inactivation [43]. The LZM activity of aquatic animals directly reflects their immunity and health status and is an important nonspecific immune index of aquatic animals [44]. Related studies have shown that NO and its oxygen metabolites are released into serum when macrophages and some non-immune cells are activated [45], which play a role in signal transduction and scavenging oxygen free radicals in the body [46]. Many studies have shown that adding BS to feed can increase the activities of LZM, SOD, ACP and AKP, and T-AOC in the serum of GIFT tilapia [25]. Similarly, CS can promote tilapia growth and improve SOD, LZM, and other enzyme activities in serum, thus enhancing the disease resistance to *Aeromonas hydrophila* [26]. Here, BS and CS could increase the activity of AKP, LZM, NO, SOD, and T-AOC in the serum of GIFT tilapia to different degrees. It has a significant effect on enhancing the nonspecific immune function of GIFT tilapia. However, the effects of KMPS and THPS on AKP, ACP, SOD, and T-AOC gradually decreased and even inhibited T-AOC with increasing use time [9]. In addition, on day 35, the content of NO in serum of tilapia in KMPS and THPS groups increased significantly. It is speculated that the frequent use of KMPS and THPS may lead to tissue damage, making the immune system produces a large amount of NO while activating macrophages and promoting histiocyte production. TNF is a class of pleiotropic cytokines that play an important role in homeostasis and disease pathogenesis. TNF-α is an important pro-inflammatory cytokine in the TNF family, which can clear the infection by activating immune cells and promoting the secretion of other cytokines [47]. Similarly, IL-1β acts as an important pro-inflammatory cytokine by recruiting more lymphocytes to colonization/invasion sites to accelerate and enhance immune effects [25]. The expression levels of IL-1β and TNF-α genes in animal bodies have similar indicative significance; upon up-regulating by a large margin, it indicates the occurrence of inflammatory reaction, and a small upward adjustment can indicate that the animal is in a high level of immune preparation [33]. Interferon-γ (IFN—γ) is a multipotent cytokine that can enhance immunity by activating signal transduction pattern recognition receptors in the congenital and adaptive immune systems [25]. It can affect cell response by regulating the expression level of multiple genes, such as improving NK cell activity and macrophage lysosome activity and promoting antigen presentation [48]. Research shows that the addition of CS to the diet could not only promote GIFT tilapia growth [26, 49] but also significantly increase the expression levels of IL-1β and IFN-γ in GIFT tilapia kidney and enhance immunity and disease resistance [50]. Similarly, BS can significantly increase the expression of complement C3, IL-1β, TNF-α, IFN-γ, and hsp-70 in the liver of tilapia and play a role in the up-regulation of the immune response [50]. In this study, BS and CS enhanced immune response and immunity by up-regulating the gene expression of TNF-α, IL-1β, and IFN-γ in the liver and spleen of GIFT tilapia. Differently, TNF-α and IL-1β of KMPS and THPS were overexpressed in the liver of tilapia, and TNF-α of KMPS was overexpressed in the spleen of tilapia, indicating that KMPS and THPS may cause inflammation or injury. Consequently, through the histomorphological observation on the 35th day of the experiment, it was found that KMPS and THPS caused different degrees of cavitation, nuclear pyknosis, and migration of liver tissue of GIFT tilapia. The results show that frequent use of KMPS and THPS could damage liver tissue of GIFT tilapia.

Generally, the growth performance, immune function, antioxidant capacity, and immune response level of tilapia are influenced by the growing water environment. The results of correlation analysis showed that COD, TOC, active phosphate, PH, nitrite, and ammonia nitrogen contents in tilapia growing water were negatively correlated to ALB serum content as well as SOD, AKP, and ACP enzyme activities. This had a certain inhibitory effect on the immunity of tilapia. On the contrary, the content of DO in the water environment can positively regulate ALB, AKP, and LZM enzyme activities in blood and IFN-γ expression level in the liver, thereby improving the immune levels of tilapia.

The above results indicate that BS and CS can mitigate tilapia growth and immune suppression caused by the increase in COD, TOC, reactive phosphate, nitrite, and ammonia nitrogen contents in the water environment. Additionally, maintaining stable pH and DO levels positively regulate the immune response of tilapia, enhance feed conversion rate, improve growth, and eventually increase the immune status of tilapia. KMPS and THPS can accelerate the degradation of harmful substances in water and improve water quality in a short period. However, due to their strong biocidal properties, prolonged usage of KMPS and THPS may disrupt the stability of the water biota environment, impacting the liver and spleen function of tilapia, there by leading to tissue damage.

## 5. Conclusion

In summary, BS and CS can effectively reduce harmful substance levels while improving water quality, growth performance, immune levels, antioxidant capacity, and immune response of tilapia. Moreover, KMPS and THPS can improve not only the water quality but also the immune response and growth performance of tilapia in the short term. However, with prolonged usage, KMPS and THPS can cause damage to water quality and negatively impact tilapia growth. Moreover, this studies have demonstrated that the escalation of deleterious substances in aquatic environments exerts an immunosuppressive impact on tilapia. Conversely, DO has been observed to possess a beneficial regulatory influence on the immune responses of tilapia.

## Author Contributions

**Conceptualization:** Liang-Gang Wang, Meng-Qian Liu, Xiao-Dong Xie, Yu-Bo Sun, Yi Zhao, Yi-Qu Ding.

**Data curation:** Liang-Gang Wang, Meng-Qian Liu, Xiao-Dong Xie, Ming-Lin Zhang, Yi Zhao, Qi Chen, Yi-Qu Ding, Zheng-Min Liang.

**Formal analysis:** Liang-Gang Wang, Meng-Qian Liu, Yu-Bo Sun, Ming-Lin Zhang, Yi Zhao, Yi-Qu Ding, Zheng-Min Liang.

**Funding acquisition:** Liang-Gang Wang, Meng-Qian Liu, Yi Zhao, Qi Chen, Mei-Ling Yu.

**Investigation:** Liang-Gang Wang, Meng-Qian Liu, Yu-Bo Sun, Yi Zhao, Qi Chen, Mei-Ling Yu.

**Methodology:** Liang-Gang Wang.

**Project administration:** Ting-Jun Hu, Wan-Wen Liang, Ying-Yi Wei.

**Resources:** Ting-Jun Hu, Wan-Wen Liang, Ying-Yi Wei.

**Software:** Liang-Gang Wang, Ting-Jun Hu, Wan-Wen Liang, Ying-Yi Wei.

**Supervision:** Ting-Jun Hu, Wan-Wen Liang, Ying-Yi Wei.

**Validation:** Liang-Gang Wang, Ming-Lin Zhang, Mei-Ling Yu, Ying-Yi Wei.

**Visualization:** Liang-Gang Wang, Ying-Yi Wei.

**Writing – original draft:** Liang-Gang Wang.

**Writing – review & editing:** Liang-Gang Wang, Meng-Qian Liu, Ting-Jun Hu, Wan-Wen Liang, Ying-Yi Wei.

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
