## [Decision Letter · Decision Letter 0]

18 May 2023

PONE-D-23-10118Effects of different water quality regulators on growth performance, immunologic function and domestic water quality of GIFT tilapia (Oreochromis niloticus, GIFT strain)PLOS ONE

Dear Dr. Ying-Yi Wei,

Thank you for submitting your manuscript to PLOS ONE. After careful consideration, we feel that it has merit but does not fully meet PLOS ONE’s publication criteria as it currently stands. Therefore, we invite you to submit a revised version of the manuscript that addresses the points raised during the review process.

We look forward to receiving your revised manuscript.

Kind regards,

Amel Mohamed El Asely

Academic Editor

PLOS ONE

Journal Requirements:

2. PLOS requires an ORCID iD for the corresponding author in Editorial Manager on papers submitted after December 6th, 2016. Please ensure that you have an ORCID iD and that it is validated in Editorial Manager. To do this, go to ‘Update my Information’ (in the upper left-hand corner of the main menu), and click on the Fetch/Validate link next to the ORCID field. This will take you to the ORCID site and allow you to create a new iD or authenticate a pre-existing iD in Editorial Manager. Please see the following video for instructions on linking an ORCID iD to your Editorial Manager account: " ext-link-type="uri" xlink:type="simple">https://www.youtube.com/watch?v=_xcclfuvtxQ"

3. Thank you for submitting the above manuscript to PLOS ONE. During our internal evaluation of the manuscript, we found significant text overlap between your submission and previous work in the [introduction, conclusion, etc.].

Please revise the manuscript to rephrase the duplicated text, cite your sources, and provide details as to how the current manuscript advances on previous work. Please note that further consideration is dependent on the submission of a manuscript that addresses these concerns about the overlap in text with published work.

[If the overlap is with the authors’ own works: Moreover, upon submission, authors must confirm that the manuscript, or any related manuscript, is not currently under consideration or accepted elsewhere. If related work has been submitted to PLOS ONE or elsewhere, authors must include a copy with the submitted article. Reviewers will be asked to comment on the overlap between related submissions (http://journals.plos.org/plosone/s/submission-guidelines#loc-related-manuscripts).]

We will carefully review your manuscript upon resubmission and further consideration of the manuscript is dependent on the text overlap being addressed in full. Please ensure that your revision is thorough as failure to address the concerns to our satisfaction may result in your submission not being considered further

4. "Thank you for stating the following in the Acknowledgments Section of your manuscript: 

"This work was financially supported by the Innovation Driven Development Fund of Guangxi [Grant number: GK AA17204081-2]. The Guangxi innovation team building project of the national modern agricultural industry technology system [Grant number: nycytxgxcxtd-14-02].We thank Guangxi University for its resources and support for this research result."

"This work was financially supported by the Innovation Driven Development Fund of Guangxi [Grant number: GK AA17204081-2]. The Guangxi innovation team building project of the national modern agricultural industry technology system [Grant number: nycytxgxcxtd-14-02].We thank Guangxi University for its resources and support for this research result."

Please include your amended state

Reviewers' comments:

Reviewer's Responses to Questions

**Comments to the Author**

1. Is the manuscript technically sound, and do the data support the conclusions?

Reviewer #1: Partly

Reviewer #2: Yes

2. Has the statistical analysis been performed appropriately and rigorously? 

Reviewer #1: Yes

Reviewer #2: Yes

3. Have the authors made all data underlying the findings in their manuscript fully available?

Reviewer #1: Yes

Reviewer #2: Yes

4. Is the manuscript presented in an intelligible fashion and written in standard English?

Reviewer #1: No

Reviewer #2: No

5. Review Comments to the Author

Reviewer #1: English language revision is a mast. Overall, the manuscript needs to be edited by someone with a solid grasp on technical writing in English.

Line 37: please replace “phenomena” by “findings.”

The conclusion of the abstract section needs to be revised.

Line 85 to 90: the aim of work needs to be rewritten.

Line 90: what are new ideas and theoretical basis which this study has provided??

Line 92: please replace “experimental anima” by “Experimental animal.”

Line 93: please replace “Oreochromisnilotcus” by “Oreochromis nilotcus”

Line 100: please replace “Experimental drug” by “Tested compounds”.

Line 122: 2.4. Feeding experiment GIFT tilapia

- What is the base for choice of the concentrations of KMPS, THPS, BS and CS that reached to the water??

- Using only one concentration of tested compounds is not sufficient to conduct a proper study. More than 2 concentrations are recommended.

- How is the amount of water evaporated from the cement ponds calculated, how is it compensated, and what are the rates of that compensation through the experimental period?

- Line 133: What is the basis for choosing the feed rate as well as the percentage of protein in the diet, even though the starting weight of the fish is 485±60 g as mentioned in line 96.

- Line 137: what is the activity that the authors recorded during the experiment?

- Line 143: (by the detection instrument.), please delete the dot.

- Line 144: 2.6. Growth performance: please delete (%) from all the formula.

- Line 162: please replace “ETDA-K2” by “K2-EDTA.” And must any abbreviation mention at the first time in the manuscript as full name.

- Line 165: EP tube, please any abbreviation must be mentioned at the first time in the manuscript as full name.

- Line 166: 3000 r/min, please correct the unit of measure.

- Line 167: please replace “glutamic oxaloacetic transaminase” by “aspartate aminotransferase” and replace “glutamic pyruvic transaminase” by “Alanine transaminase”.

- Line 173: 2.8. Cytokines gene expression: please clarify number of liver and spleen samples that had been collected/group.

- Table 2: Please make sure that the accession number of IFN-γ is correct.

- Line 192 2.10. Data analysis: This part does not contain Data normality and homogeneity.

- Figure 1:

• resolution very poor, besides the significant letter are very confused. It is preferable to convert this figure to a table.

• Please unify the meaning of significant letter. For example, at figure 1C the letter “a” for the lowest value except at the 5th week results, it was added to the highest value???

- Line 211: please delete “blank”.

- Table 3:

• Initial weight in the table was ranged from 475.50 to 479.55, while at the line 96, the authors mentioned that it was 485±60 g. Please revise.

- Line 280: please delete “blank”.

- Line 294: The method of presentation of histopathological findings is inadequate, unclear, and disorganized. Please rewrite this part.

- Why did the authors not apply histopathological exam to kidney tissue to confirm the disturbance in renal function??

- Line 303: please clarify how the correlation between water environmental factors and immune factors was statistically applied in the data analysis section.

- The authors at line 303 mentioned that the correlation was applied between water environmental factors and immune factors, but creatinine, urea, ALT, ...... are not from the immune factors???

- Line 311: please replace “Discuss” by “Discussion.”

- Line 347 to 351: please delete.

- The authors do not clarify the attribution for the growth performance enhancement by the addition of BS and CS in water.

- Line 359 to 363: please delete.

- Line 366 to 369: please delete.

- Discussion:

• The discussion section must be displayed in the same order as the results section.

• There is no explanation for a lot of results.

• The outcomes of the various types of analysis should be connected in the discussion section.

- Conclusion: The section should not be elaborated.

Reviewer #2: In the present manuscript entitled: “Effects of different water quality regulators on growth performance, immunologic function and domestic water quality of GIFT tilapia (Oreochromis niloticus, GIFT strain)”, authors studied the effect of some water quality regulators such as potassium monopersulfate, tetrakis hydroxymethyl phosphonium sulfate, bacillus subtilis and chitosan on water quality, growth performance, physiological, and immune responses of fish. The study provides some important data which will be of value for fish farmers. The manuscript requires massive editing, rephrasing, and linguistic improvement. There are many wordy and incomplete sentences. My decision is accepting after minor revision

Abstract

Line 16 delete “and” replace with,

Line 28 delete “blank”

Line 33 etc???? what does you mean…write the name of cells

Line 33-36 …complete the sentence…which groups do you mean?

Line 37-38 mention exactly which groups exhibited these pathological changes

Line 39 delete “improve the growth performance” delete improve

Line 40 delete level

Line 42-44 rephrase and clarify “However, the functions of liver, spleen and kidney of tilapia were affected with the increase of use time, resulting in inflammatory reaction and liver tissue damage, affecting the growth of GIFT tilapia”.

Introduction

Line 56- 59 write the sentence in the present tense not in the past

Material and methods

How did you confirm that experimental fish are healthy?

Why did you choose these specific drug concentrations? 1.5mg/L, 1mg/L, 5mg/L and 2mg/L?

Line 92 Experimental anima?

Line 107 The experimental basic feed uses??????? Correct the sentence

Line 145 Growth performance parameters were

Line 184 histopathology

Line 199-208 rephrase and clarify.

Results

The results section need more clarification, express the findings more succinctly

Discussion

Discussion section is redundant need more clarification

Line 311 discuss…………..correct

-“At present, the deterioration of water quality caused by environmental

pollution has become the main reason for the breeding of aquatic animal diseases”what do you mean?………… this sentence is not correct

- “Drug prevention can effectively……….” what do you mean?………… this sentence is not correct

Conclusion

Conclusion need more clarification

-“With the increase of use time, it will not only cause water quality damage, but also

damage the liver, spleen and kidney function” what do you mean by “it”

Figures

-The ID of the pathological figures should be written more legibly.

6. PLOS authors have the option to publish the peer review history of their article (what does this mean?). If published, this will include your full peer review and any attached files.

Reviewer #1: No

Reviewer #2: No

---

## [Author Response · Author response to Decision Letter 0]

21 Jul 2023

Response to Reviewers

Reviewer #1: English language revision is a mast. Overall, the manuscript needs to be edited by someone with a solid grasp on technical writing in English.

Response: Thank you for your suggestions. We apologize for the poor language of our manuscript. We worked on the manuscript for a long time and the repeated addition and removal of sentences and sections obviously led to poor readability. We have now worked on both language and readability and have also involved native English speakers for language corrections (as shown in Annex 1). We really hope that the flow and language level have been substantially improved. 

Question 1. Line 37: please replace "phenomena" by "findings." 

Answer: This part of the content has been modified in the manuscript.

Question 2. The conclusion of the abstract section needs to be revised.

Answer: The abstract section has been revised and the conclusion of the abstract section showed on lines 37-42 of the revised manuscript.

Line 37-42:In conclusion, these four water quality regulators, mainly BS and CS, could improve the growth performance and immunity of GIFT tilapia to varying degrees by regulating the water quality and then further increasing the expression levels of immune-related factors or the activity of antioxidant-related enzymes of GIFT tilapia. On the contrary, the prolonged use of KMPS and THPS may gradually diminish their growth-enhancing properties and potentially hinder the growth of GIFT tilapia.

Question 3. Line 85 to 90: the aim of work needs to be rewritten. 

Answer:The aim of the work has been rewritten in lines 81 to 85 of the revised manuscript.

Line 81 to 85:Consequently, this study aims to comprehensively evaluate the effects of KMPS, THPS, BS and CS on GIFT tilapia by determining the growth performance, blood physiology and biochemistry, histopathology, antioxidant enzymatic activity, and immunity-related factors. Eventually, our results may provide a theoretical basis for selecting and using the best effective water quality regulators in aquaculture. 

Question 4. Line 90: what are new ideas and theoretical basis which this study has provided?? 

Answer:The purpose of the experiment is to provide a theoretical basis for the selection and use of the best water quality regulators in tilapia culture.

Line 83 to 85: Eventually, our results may provide a theoretical basis for selecting and using the best effective water quality regulators in aquaculture. 

Question 5. Line 92: please replace "experimental anima" by "Experimental animal." 

Answer: The "experimental anima" have been replaced by "Experimental animal" in line 90 of the manuscript.

Question 6. Line 93: please replace "Oreochromisnilotcus" by "Oreochromis nilotcus" 

Answer: The content has been deleted in the new manuscript.

Question 7. Line 100: please replace "Experimental drug" by "Tested compounds". 

Answer: The "Experimental drug" have been replaced by "Tested compounds" in line 96 of the manuscript.

Question 8. Line 122: 2.4. Feeding experiment GIFT tilapia 

- What is the base for choice of the concentrations of KMPS, THPS, BS and CS that reached to the water?? 

Answer: KMPS, THPS, BS, and CS are all commercial formulations, and the concentrations used in the experiment are the optimal clinically recommended doses provided in the respective product manuals.

Question 9.- Using only one concentration of tested compounds is not sufficient to conduct a proper study. More than 2 concentrations are recommended. 

Answer: In the experiment, each drug concentration is the best clinical recommended dose in the instruction manual. A corresponding supplementary explanation has been made in the manuscript .

Line 120 to 124: With the exception of the control group, the experimental cement ponds were treated with an aqueous solution containing KMPS, THPS, BS, and CS. This treatment was administered once every seven days, resulting in concentrations of 1.5, 1, 5, and 2 mg/L respectively. These concentrations were chosen based on the recommended clinical dosage provided in the instructions for each preparation.

Question 10. - How is the amount of water evaporated from the cement ponds calculated, how is it compensated, and what are the rates of that compensation through the experimental period? 

Answer: At the beginning of the experiment, the water level of each pool is marked, and the evaporated water is added in time to maintain the amount of water in the experiment.

Line 124 to 127: The water in the experimental cement ponds remained unchanged throughout the experiment. It was refilled as necessary to compensate for water loss due to evaporation, ensuring that the water volume remained constant following the initial water level markings.

Question 11. - Line 133: What is the basis for choosing the feed rate as well as the percentage of protein in the diet, even though the starting weight of the fish is 485±60 g as mentioned in line 96. 

Answer: (1)The food intake of adult tilapia is relatively stable, and the efficiency of digestion and absorption of food is also high. Generally speaking, the daily feed weight of 1% to 3% is used as the reference range, and the specific rate needs to be adjusted according to the actual situation.

(2)The protein content in feed is determined with reference to the protein content in commercial feed in the market. For example, the protein content in the feed formula of tilapia in Haida Group is more than 40%. In addition, the study of Mohsen et al[1] showed that the growth performance of Nile tilapia was the best when 45%CP was added to the feed and the culture density was 150ind / m3.

[1]Mohsen Abdel-Tawwab. Effects of dietary protein levels and rearing density on growth performance and stress response of Nile tilapia, oreochromis niloticus(L.). International Aquatic Research(1). 2012. doi:https://doi.org/10.1186/2008-6970-4-3.

Question 12.- Line 137: what is the activity that the authors recorded during the experiment?

Answer: The growth and survival of tilapia were observed. The corresponding content has been added to "3.2. Growth performance " in line 208 to 209 of the manuscript.

Line 207 to 208: During the experiment, tilapia had no obvious pathological changes and abnormal death, and its activity, food intake and body color were normal. 

Question 13. - Line 143: (by the detection instrument.), please delete the dot. 

Answer: The content has been deleted or replaced in the manuscript 

Question14. - Line 144: 2.6. Growth performance: please delete (%) from all the formula. 

Answer: The (%) from all the formula in line 144 to 149 has been deleted.

Question 15. - Line 162: please replace "ETDA-K2" by "K2-EDTA." And must any abbreviation mention at the first time in the manuscript as full name.

Answer: This error has been corrected in the manuscript , and the same type of problems have been checked and corrected in full.

Line 159: dipotassium ethylenediamine tetraacetate(K2-ETDA)

Question 16. - Line 165: EP tube, please any abbreviation must be mentioned at the first time in the manuscript as full name. 

Answer: It has been supplemented and corrected in this article as "Eppendorf tube", and the same type of problems have been checked and corrected in full.

Line 161 : Eppendorf tube

Question 17. - Line 166: 3000 r/min, please correct the unit of measure. 

Answer: 3000 r/min has been converted to the international common unit 956 rcf in line 162 of the manuscript..

Line 162 : with an rcf of 956.

Question 18. - Line 167: please replace "glutamic oxaloacetic transaminase" by "aspartate aminotransferase" and replace "glutamic pyruvic transaminase" by "Alanine transaminase". 

Answer: The "glutamic oxaloacetic transaminase" have been replaced by "aspartate aminotransferase" and the "glutamic pyruvic transaminase" have been replaced by "alanine transaminase"in line 164 of the manuscript.

Question 19. - Line 173: 2.8. Cytokines gene expression: please clarify number of liver and spleen samples that had been collected/group. 

Answer: Additional notes have been made in line 171 to 175 of the manuscript.

Line 171 to 175: Four liver or spleen tissue samples were taken from each replicate/group (n = 12 tilapias/group). The samples were ground and preserved with RNA Keeper tissue stabilizer (Vazyme, China) to determine cytokine. The study employed the Trizol reagent to extract the total RNA of liver or spleen tissues, followed by detecting the RNA purity utilizing 1.5% agarose gel electrophoresis and then determining the total RNA concentration of the extracted samples. 

Question 20. - Table 2: Please make sure that the accession number of IFN-γ is correct.

Answer: The accession number of IFN- γ has been corrected in Table 2: NM_001287402.1

Question 21. - Line 192 2.10. Data analysis: This part does not contain Data normality and homogeneity.

Answer: The methods of data analysis have been elaborated and supplemented in the manuscript.

Line 191 to 195: Statistical comparisons of experimental data were performed by one-way analysis of variance (ANOVA) using SPSS 22.0 software (IBM, USA). Duncan's Multiple Range test was used to identify significant differences. Data are presented as mean ± standard error. Lowercase letters (a, b, c, d, and e) denote significant differences between different sampling groups (determined by Duncan's test, P 0.05).

- Figure 1:

Question 22. • resolution very poor, besides the significant letter are very confused. It is preferable to convert this figure to a table.

Answer: Because the amount of data is very large, making a table may be very large, so we still use the form of a graph. The chart can also better show the changing trend of water quality of each treatment group over time, which can not be expressed intuitively in the table. In addition, we have modified the resolution and letter marking of the image. Of course, we have also added data tables here, but the volume is large and it is difficult to see the trend intuitively. If the table can achieve better results, then we will add it to the manuscript in the next revision.

Table:

 Times / week Control KMPS THPS BS CS

COD 1 5.51±0.56b 4.70±0.17a 5.33±0.48b 5.37±0.56b 5.07±0.52ab

 2 6.77±0.28bc 5.65±0.56a 7.07±0.66c 6.22±0.46b 6.64±0.39bc

 3 7.76±0.28c 6.44±0.19a 7.09±0.46b 6.14±0.45a 7.54±0.44c

 4 8.18±0.48b 7.88±0.31b 8.02±0.33b 7.01±0.99a 8.13±0.35b

 5 9.60±0.32c 9.36±0.26c 8.86±0.63b 7.33±0.37a 9.18±0.37ab

DO 1 8.26±0.45a 8.26±0.32a 8.29±0.51a 8.01±0.24a 8.12±0.44a

 2 7.53±0.16b 7.63±0.14bc 7.17±0.36a 7.48±0.11b 7.89±0.35c

 3 7.27±0.23a 7.48±0.35a 7.24±0.22a 7.90±0.09b 7.95±0.31b

 4 6.99±0.23a 7.07±0.21a 6.93±0.14a 8.01±0.10c 7.62±0.18b

 5 6.94±0.15a 7.26±0.36a 7.21±0.41a 8.02±0.38b 7.82±0.33b

pH 1 7.76±0.21b 7.53±0.21a 7.76±0.14b 7.50±0.24a 7.65±0.11ab

 2 8.17±0.17c 7.92±0.22bc 7.76±0.44ab 7.54±0.38a 7.86±0.19abc

 3 8.53±0.08b 7.97±0.37a 8.89±0.37c 7.78±0.24a 8.31±0.23b

 4 8.89±0.13d 7.56±0.19a 9.07±0.19e 8.03±0.15b 8.59±0.03c

 5 9.65±0.22b 8.62±0.74a 8.68±0.28a 8.25±0.25a 8.66±0.22a

TOC 1 4.43±0.61ab 3.94±0.36a 4.70±0.32b 4.46±0.42ab 4.31±0.42ab

 2 4.71±0.20a 4.79±0.36a 5.30±0.45bc 4.94±0.24ab 5.50±0.58c

 3 5.79±0.40c 4.95±0.45a 5.35±0.26b 5.25±0.26ab 5.77±0.26c

 4 6.62±0.16b 5.72±0.42a 6.39±0.32b 6.31±0.23b 6.27±0.32b

 5 7.60±0.38c 7.22±0.35b 6.93±0.20ab 6.78±0.33a 7.17±0.21b

Ammonia nitrogen

 1 1.67±0.42a 1.44±0.46a 1.69±0.46a 1.62±0.33a 1.47±0.41a

 2 2.60±0.19b 2.54±0.10b 2.14±0.17a 2.28±0.04a 2.45±0.21b

 3 3.34±0.20d 2.76±0.10bc 2.89±0.45c 2.20±0.12a 2.61±0.09b

 4 3.76±0.05d 2.86±0.06b 3.24±0.12c 2.55±0.06a 2.85±0.13b

 5 4.39±0.25d 3.10±0.18b 3.61±0.24c 2.55±0.18a 3.20±0.11b

Phosphate

 1 0.33±0.06a 0.33±0.07a 0.31±0.34a 0.34±0.03a 0.30±0.07a

 2 0.47±0.02c 0.38±0.03a 0.37±0.03a 0.39±0.03ab 0.42±0.02b

 3 0.62±0.03b 0.49±0.02a 0.50±0.03a 0.48±0.03a 0.50±0.02a

 4 0.73±0.02b 0.57±0.03a 0.57±0.02a 0.56±0.02a 0.58±0.03a

 5 0.85±0.04d 0.73±0.04b 0.68±0.02ab 0.71±0.03a 0.77±0.04c

Nitrite

 1 25.57±4.20b 28.57±6.55b 26.29±3.55b 18.71±4.19a 24.43±4.61b

 2 36.20±1.34c 35.35±1.48bc 33.82±2.80b 26.88±0.92a 36.19±1.87c

 3 45.25±6.40b 34.64±1.80a 35.79±5.24a 34.43±2.88a 38.36±1.97a

 4 56.33±2.49e 37.67±1.49b 51.62±3.51d 33.81±2.17a 44.38±2.14c

 5 65.33±2.56d 45.43±4.04b 58.25±3.43c 41.00±3.56a 44.46±3.16ab

Question 23.• Please unify the meaning of significant letter. For example, at figure 1C the letter "a" for the lowest value except at the 5th week results, it was added to the highest value??? 

Answer: This error has been corrected in Fig.1, and the rest of the images have been checked for this problem.

Fig.1:

Question 24. - Line 211: please delete "blank". 

Answer: Corresponding amendments have been made in the manuscript.

- Table 3:

Question 25. • Initial weight in the table was ranged from 475.50 to 479.55, while at the line 96, the authors mentioned that it was 485±60 g. Please revise. 

Answer: The 485 ±60 g has been corrected to 475.50~479.55 g in line 91 of the manuscript.

Question 26. - Line 280: please delete "blank".

Answer: Corresponding amendments have been made in the manuscript.

Question 27. - Line 294: The method of presentation of histopathological findings is inadequate, unclear, and disorganized. Please rewrite this part. 

Answer:This part of the content has been reorganized and improved in lines 278 to 284 of the manuscript.

Line 278 to 284 : No significant histopathological alterations were observed in the liver or spleen tissue of the control, BS, and CS groups on day 35. The tissue structure of the liver or spleen was clear, and the cells were evenly distributed, arranged orderly, intact, and with full nuclei (Figs. 4a, d–e and 5 a, d– e). Nevertheless, the liver of tilapia in KMPS and THPS groups showed significant pathological changes, such as tissue vacuolation to varying degrees, cell shrinkage, cell nucleus shift, and nucleolar pyknosis (Figs. 4b–c). Nevertheless, the spleen showed no significant histopathological changes (Figs. 5b–c).

Question28. - Why did the authors not apply histopathological exam to kidney tissue to confirm the disturbance in renal function?? 

Answer: This article mainly focuses on the immune changes, so there is no pathological examination of the kidney.

Question 29. - Line 303: please clarify how the correlation between water environmental factors and immune factors was statistically applied in the data analysis section. 

Answer: The statistical application of the correlation between water environmental factors and immune factors in the part of data analysis has been supplemented in line 288 to 289 of the manuscript.

Line 288 to 289: The Pearson correlation between water environment factors and immune-related factors of tilapia was analyzed by the omicshare cloud tool (https://www.omicshare.com/tools/).

Question 30. - The authors at line 303 mentioned that the correlation was applied between water environmental factors and immune factors, but creatinine, urea, ALT, ...... are not from the immune factors??? 

Answer: Creatinine, urea, ALT and AST are true that these factors are not immune factors, so we redescribe and analyze them in lines 296 to 299 of the manuscript..

Line 296 to 299: In addition, we analyzed the correlation between environmental factors and blood biochemistry (Fig.6). The results showed that even though the blood biochemistry of tilapia changed greatly during the experiment, the correlation between environmental factors and the contents of CRE, BUN, AST and ALT in tilapia blood was not significant(P ＞ 0.05).

Question 31. - Line 311: please replace "Discuss" by "Discussion." 

Answer: The "Discuss" have been replaced by "Discussion" in line 301 of the manuscript.

Question 32. - Line 347 to 351: please delete. 

Answer: Corresponding amendments have been made in the manuscript.

Question 33. - The authors do not clarify the attribution for the growth performance enhancement by the addition of BS and CS in water.

Answer: The relevant discussion is supplemented in line 329 to 332 of the manuscript.

Line 329 to 332: According to the findings of Liu et al., bacillus subtilis can provide a better intestinal flora environment for tilapia as well as improve intestinal absorption and metabolism of nutrients to promote tilapia growth [2]. Similarly, CS can promote tilapia growth by increasing feed conversion rate [3, 4].

[2].Liu Q, LutingPan, XianhuiHuang, YinDu, XuesongQin, JunqiZhou, KangqiWei, ZinaChen, ZhongMa, HuaweiHu, TingjunLin, Yong. Dietary supplementation of Bacillus subtilis and Enterococcus faecalis can effectively improve the growth performance, immunity, and resistance of tilapia against Streptococcus agalactiae. Aquaculture Nutrition. 2021;27(4). doi: https://doi.org/10.1111/ANU.13256.

[3].Wu S. The growth performance, body composition and nonspecific immunity of Tilapia ( Oreochromis niloticus ) affected by chitosan. International Journal of Biological Macromolecules. 2020;145:682-5. doi: https://doi.org/10.1016/j.ijbiomac.2019.12.235.

[4].Rei A, Saa B, Kyf C, Ag D, Aia E, Ea F, et al. The effects of chitosan-vitamin C nanocomposite supplementation on the growth performance, antioxidant status, immune response, and disease resistance of Nile tilapia ( Oreochromis niloticus ) fingerlings. Aquaculture. 2020;534. doi: https://doi.org/10.1016/J.AQUACULTURE.2020.736269.

Question 34. - Line 359 to 363: please delete. 

Answer: We have deleted this part of the content as required, and made additions and adjustments to the relevant contents of the manuscript.

Line 327 to 332: Herein, BS and CS can significantly improve the growth performance of GIFT tilapia, reduce CF, and contribute significantly to growth enhancement. According to the findings of Liu et al., bacillus subtilis can provide a better intestinal flora environment for tilapia as well as improve intestinal absorption and metabolism of nutrients to promote tilapia growth . Similarly, CS can promote tilapia growth by increasing feed conversion rate .

Question 35. - Line 366 to 369: please delete. 

Answer: We have deleted this part of the content as required, and made additions and adjustments to the relevant contents of the manuscript.

Line 337 to 343: In our experiment, the CRE and BUN serum content of KMPS and THPS-treated GIFT tilapia increased significantly, which may be resulted from the metabolic disorders caused by renal tissue damage . The results showed that KMPS and THPS had certain effects on the renal function of GIFT tilapia. In our experiment, KMPS and THPS increased ALT and AST serum contents of GIFT tilapia, suggesting that these two water quality regulators have certain side effects on the liver and kidneys of GIFT tilapia.

- Discussion: 

Question 36. • The discussion section must be displayed in the same order as the results section. 

Answer: According to the order in the discussion, we compare "3.3. Blood biochemical" and "3. 4. Blood physiology" changed places.The content is shown in 222 to 246 line of the manuscript

Question 37. • There is no explanation for a lot of results. 

Answer: A part of the results in the study showed insignificant changes after medication, therefore, detailed analysis was not conducted. Additionally, further discussion and analysis of the discussed results were also included in the manuscript

Question 38. • The outcomes of the various types of analysis should be connected in the discussion section. 

Answer: We have strengthened the connections between various analysis results in the discussion section of the manuscript. Furthermore, we have also conducted a comprehensive analysis of the experimental results.

Line 434 to 442: The above results indicate that BS and CS can mitigate tilapia growth and immune suppression caused by the increase in COD, TOC, reactive phosphate, nitrite, and ammonia nitrogen contents in the water environment. Additionally, maintaining stable pH and DO levels positively regulate the immune response of tilapia, enhance feed conversion rate, improve growth, and eventually increase the immune status of tilapia. KMPS and THPS can accelerate the degradation of harmful substances in water and improve water quality in a short period. However, due to their strong biocidal properties, prolonged usage of KMPS and THPS may disrupt the stability of the water biota environment, impacting the liver and spleen function of tilapia, there by leading to tissue damage.

Question 39. - Conclusion: The section should not be elaborated.

Answer: The conclusion has been restated and sorted out in the manuscript.

Line 444 to 451: In summary, BS and CS can effectively reduce harmful substance levels while improving water quality, growth performance, immune levels, antioxidant capacity, and immune response of tilapia. Moreover, KMPS and THPS can improve not only the water quality but also the immune response and growth performance of tilapia in the short term. However, with prolonged usage, KMPS and THPS can cause damage to water quality and negatively impact tilapia growth. Moreover, this studies have demonstrated that the escalation of deleterious substances in aquatic environments exerts an immunosuppressive impact on tilapia. Conversely, DO has been observed to possess a beneficial regulatory influence on the immune responses of tilapia.

Reviewer #2: In the present manuscript entitled: "Effects of different water quality regulators on growth performance, immunologic function and domestic water quality of GIFT tilapia (Oreochromis niloticus, GIFT strain)", authors studied the effect of some water quality regulators such as potassium monopersulfate, tetrakis hydroxymethyl phosphonium sulfate, bacillus subtilis and chitosan on water quality, growth performance, physiological, and immune responses of fish. The study provides some important data which will be of value for fish farmers. The manuscript requires massive editing, rephrasing, and linguistic improvement. There are many wordy and incomplete sentences. My decision is accepting after minor revision.

Response: We thank the reviewers for their affirmation of the content of the study.We have polished our manuscript carefully and corrected the grammatical, styling, and typos found in our manuscript. The amendment certificate is shown in Annex 1.

Question 1. Line 28 delete"blank"

Answer: Corresponding amendments have been made in the manuscript.

Question 2. Line 16 delete "and" replace with, 

Answer: The "and" have been replaced by "," in line 16 of the manuscript.

Question 3. Line 28 delete "blank" 

Answer: Corresponding amendments have been made in the manuscript.

Question 4. Line 33 etc???? what does you mean…write the name of cells 

Answer: This part has been redescribed in the manuscript

Question 5. Line 33-36 …complete the sentence…which groups do you mean? 

Answer: These contents have been redescribed in the manuscript.

Question 6. Line 37-38 mention exactly which groups exhibited these pathological changes 

Answer: These contents have been redescribed in the manuscript.

Question 7. Line 39 delete "improve the growth performance" delete improve 

Answer: These contents have been redescribed in the manuscript.

Question 8. Line 40 delete level 

Answer: These contents have been redescribed in the manuscript.

Question 9. Line 42-44 rephrase and clarify "However, the functions of liver, spleen and kidney of tilapia were affected with the increase of use time, resulting in inflammatory reaction and liver tissue damage, affecting the growth of GIFT tilapia".

Answer: These contents have been redescribed in the manuscript.

Question 3~9:

Line 19 to 42:

 The findings indicated that implementing the four water quality regulators resulted in a decrease in the content of ammonia nitrogen, active phosphate, nitrite, total organic carbon (TOC), and chemical oxygen demand (COD) in the water. Additionally, these regulators were found to maintain dissolved oxygen (DO) levels and pH of the water effectively. Furthermore, using these regulators demonstrated positive effects on various physiological parameters of GIFT tilapia, including improvements in final body weight, weight gain rate (WGR), specific growth rate (SGR), condition factor (CF), feed conversion ratio (FCR), spleen index (SI), hepato-somatic index (HSI), immune cell count, the activity of antioxidant-related enzymes (Nitric oxide, NO and Superoxide dismutase, SOD), and mRNA expression levels of immunity-related factors (Tumor Necrosis Factor-alpha, TNF-α and Interleukin-1 beta, IL-1β) in the liver and spleen. Notably, the most significant improvements were observed in the groups treated with the BS and CS water quality regulators. Moreover, BS and CS groups exhibited significantly higher serum levels of albumin (ALB) and total protein (TP) (P 0.05), whereas the other indicators showed no significant difference (P 0.05) compared to the control group. However, the KMPS and THPS groups of GIFT tilapia exhibited significantly higher serum levels of aspartate aminotransferase (AST), alanine transaminase (ALT), creatinine (CRE) and blood urea nitrogen (BUN) (P 0.05), whereas they exhibited significantly decreased HSI (P 0.05). In addition, the partially pathological observations revealed the presence of cell vacuolation, nuclear shrinkage, and pyknosis within the liver. In conclusion, these four water quality regulators, mainly BS and CS, could improve the growth performance and immunity of GIFT tilapia to varying degrees by regulating the water quality and then further increasing the expression levels of immune-related factors or the activity of antioxidant-related enzymes of GIFT tilapia. On the contrary, the prolonged use of KMPS and THPS may gradually diminish their growth-enhancing properties and potentially hinder the growth of GIFT tilapia.

Question 10. How did you confirm that experimental fish are healthy?

Answer: The "healthy" is difficult to define. We cannot guarantee that every fish selected for the experiment has a normal nutritional status and is completely disease-free. However, we can ensure that the experimental fish do not carry specific pathogens or exhibit obvious injuries and show no abnormal symptoms based on clinical observations. We have revised the relevant statements in the manuscript to avoid any misunderstandings.

Line 91 to 95: Totally, 1500 GIFT tilapia weighing 475.50 ± 4.05 g were collected from the National Guangxi Nanning Tilapia Seed Farm of Guangxi Fisheries Research Institute and were adaptively fed for one week. The health status of the experimental fish and the water quality parameters were monitored daily, confirming that no abnormal symptoms were observed during clinical observations.

Question 11. Why did you choose these specific drug concentrations? 1.5mg/L, 1mg/L, 5mg/L and 2mg/L? 

Answer: In the experiment, each drug concentration is the best clinical recommended dose in the instruction manual. A corresponding supplementary explanation has been made in the manuscript .

Line 123 to 124: These concentrations were chosen based on the recommended clinical dosage provided in the instructions for each preparation.

Question 12. Line 92 Experimental anima? 

Answer: The "Experimental anima" has been replaced with "Experimental animal" in line 90 the manuscript.

Question 13. Line 107 The experimental basic feed uses??????? Correct the sentence 

Answer: The sentence has been rephrased and corrected in line 98 to 100 the manuscript.

Line 103 to 105: The experimental diets included five main raw materials of basic feed, such as fish meal, soybean meal, rapeseed meal, cellulose, and flour. Table 1 lists the raw material composition and nutrition level of these experimental diets. 

Question 14. Line 145 Growth performance parameters were 

Answer: The sentence has been corrected in line 140 to 143 the manuscript

Line 141 to 143 : The growth performance parameters were evaluated using weight gain rate (WGR (1)), specific growth rate (SGR(2)), condition factor (CF (3)), feed conversion ratio (FCR(4)), spleen index (SI (5)), hepato-somatic index (HSI(6)). The calculation formula utilized was as follows.

Question 15. Line 184 histopathology 

Answer: The "Histomorphology" has been replaced with "Histopathology" in line 183 the manuscript.

Results 

Question 16. Line 199-208 rephrase and clarify. 

Answer: The section has been rephrased and clarified in line 198 to 204 the manuscript. 

Line 198 to 204: Fig. 1 shows the effects of the four water quality regulators on the growth water quality parameters of tilapia. The results showed that KMPS, THPS, BS, and CS could reduce the water contents of ammonia nitrogen, active phosphate, nitrite, TOC, and COD to different degrees, particularly BS, which had the most significant effect (Figs. 1c–g). Meanwhile, BS and CS significantly maintained the stability of DO and pH of the water environment (Figs. 1a–b). Although KMPS and THPS had not significantly maintained the stability of water DO, KMPS could effectively stabilize water pH (Fig. 1b). 

Question 17. The results section need more clarification, express the findings more succinctly

Answer: The results section of the manuscript has undergone a reorganization and refinement.

Line 207 to 217: During the experiment, tilapia had no obvious pathological changes and abnormal death, and its activity, food intake and body color were normal. Table 3 demonstrates the effects of the four water quality regulators on the growth performance of GIFT tilapia. On day 18, the four water quality regulators groups exhibited significantly increased FBW, WGR, CF, and SGR of GIFT tilapia (P 0.05) compared to the control group. Meanwhile, the THPS, BS, and CS groups exhibited significantly decreased FCR (P 0.05). Additionally, the KMPS and THPS groups exhibited significantly decreased HSI (P 0.05). On day 35, the FBW of tilapia in the four water quality regulator groups increased significantly (P 0.05). At the same time, the BS and CS groups showed significantly increased WGR, SGR, SI, and HSI of tilapia (P 0.05) while showing significantly decreased FCR (P 0.05). The KMPS and THPS groups showed a significant decrease in the HSI of tilapia (P 0.05).

Line 248 to 257: Table 6 indicates the effects of the four water quality regulators on the immune function and antioxidant capacity of GIFT tilapia. After continuous administration for 18 days, AKP, SOD, and T-AOC levels in the four different water quality regulator groups were significantly higher than in the control (P 0.05). Additionally, the T-AOC level in the BS group was significantly higher than in the KMPS or CS group (P 0.05). The effects of each experimental group on LZM and NO showed no significant difference (P 0.05).

On day 35, BS and CS significantly increased ACP, AKP, NO, LZM, SOD, and T-AOC levels (P 0.05) compared to the control group. Moreover, the KMPS and THPS significantly increased SOD and NO levels (P 0.05) while significantly decreasing T-AOC serum levels, with KMPS exhibiting the most significant difference (P 0.05).

Line 262 to 272: Figs. 2–3 show the mRNA expression levels of TNF-α, IL-1β, and IFN-γ of the liver or spleen are shown in . On day 18, the TNF-α, IL-1β, and IFN-γ mRNA expression levels in the BS and CS groups were significantly higher than in the control group (P 0.05). Additionally, the TNF-α in the BS and CS groups was significantly higher than in the control group (P 0.05).

On day 35, The mRNA expression levels of TNF-α, IL-1β, and IFN-γ of the liver or spleen in the BS and CS groups exhibited significantly higher levels than in the control group (P 0.05). Furthermore, the TNF-α and IL-1β levels of the liver in the KMPS and THPS groups were significantly higher than in all other experimental groups (P 0.05). Similarly, the TNF-α and IL-1β levels of the spleen in the KMPS group were significantly higher than in all other experimental groups (P 0.05). The TNF-α level of the spleen in the THPS group was significantly higher than in the control group (P 0.05).

Line 278 to 284: No significant histopathological alterations were observed in the liver or spleen tissue of the control, BS, and CS groups on day 35. The tissue structure of the liver or spleen was clear, and the cells were evenly distributed, arranged orderly, intact, and with full nuclei (Figs. 4a, d–e and 5 a, d– e). Nevertheless, the liver of tilapia in KMPS and THPS groups showed significant pathological changes, such as tissue vacuolation to varying degrees, cell shrinkage, cell nucleus shift, and nucleolar pyknosis (Figs. 4b–c). Nevertheless, the spleen showed no significant histopathological changes (Figs. 5b–c).

Line 288 to 299: The Pearson correlation between water environment factors and immune-related factors of tilapia was analyzed by the omicshare cloud tool (https://www.omicshare.com/tools/). The results showed that the levels of COD, TOC, active phosphate, pH, nitrite, and ammonia nitrogen in GIFT tilapia growing water environment were positively correlated to T-AOC in serum while negatively correlated to ALB and SOD (P 0.05; Fig. 6). Active phosphate and ammonia nitrogen were significantly negatively correlated to AKP and ACP (P 0.05). A significant positive correlation was found between DO and ALB, AKP, LZM, and IFN- γ in the liver (P 0.05).

 Additionally, we analyzed the correlation between environmental factors and blood biochemistry (Fig.6). The results showed that even though the blood biochemistry of tilapia changed greatly during the experiment, the correlation between environmental factors and the contents of CRE, BUN, AST and ALT in tilapia blood was not significant (P ＞ 0.05).

Discussion 

Question 18. Discussion section is redundant need more clarification 

Answer: Some parts of the manuscript have been modified and adjusted.

The parts of the manuscript have been deleted: In general, water quality is the key to the success or failure of aquaculture. Currently, the deterioration of water quality has become a major cause for the proliferation and transmission of diseases among aquatic animals. Although there are many drugs available that can effectively prevent the occurrence of diseases, it is difficult to address this problem at its root.The development of water quality regulator is a valuable solution in sustainable aquaculture.

Line 302 to 307 has been rephrased and clarified: The growth rate and immune level of tilapia in aquaculture are significantly influenced by the quality of water. Previous research has indicated that ammonia, nitrogen, and nitrite are significant toxic substances that arise from the feces and residual feed of aquatic animals during aquaculture. These substances can potentially result in the mortality of fish and shrimp, particularly when their concentrations exceed certain thresholds[19, 20]. As a result, reducing these deleterious substances has emerged as the central objective and challenge in water quality regulation.

Line 327 to 333 has been rephrased and clarified: Herein, BS and CS can significantly improve the growth performance of GIFT tilapia, reduce CF, and contribute significantly to growth enhancement. According to the findings of Liu et al., bacillus subtilis can provide a better intestinal flora environment for tilapia as well as improve intestinal absorption and metabolism of nutrients to promote tilapia growth . Similarly, CS can promote tilapia growth by increasing feed conversion rate . Currently, there is a lack of reported literature on applying KMPS and THPS in tilapia culture.

Line 335 to 343 has been rephrased and clarified: Herein, BS and CS can significantly improve the growth performance of GIFT tilapia, reduce CF, and contribute significantly to growth enhancement. According to the findings of Liu et al., bacillus subtilis can provide a better intestinal flora environment for tilapia as well as improve intestinal absorption and metabolism of nutrients to promote tilapia growth . Similarly, CS can promote tilapia growth by increasing feed conversion rate . Currently, there is a lack of reported literature on applying KMPS and THPS in tilapia culture.

Line 361 to 362 has been rephrased and clarified: Immune cells exhibit a close association with immune response, growth, and metabolism, while also serving as a partial indicator of immune function potency .

Line 370 to 376 has been rephrased and clarified: The difference is that the total number of leukocytes, lymphocytes, intermediate cells and neutrophils in the blood of tilapia in KMPS and THPS groups is abnormally higher than the normal value, suggesting that there may be inflammatory damage in the body. Humoral responses involve several unspecific enzymes or factors, such as lysozyme, superoxide, and dismutase, among others. These components function to eradicate pathogens either through direct pathogen killing or by impeding pathogen growth and dissemination.

Line 377 to 383 has been rephrased and clarified: AKP is a key enzyme in metabolism and antioxidation, which participates in the regulation of phosphate groups in the body . Similarly, ACP is an important enzyme in material metabolism and signal transduction and plays an important role in the metabolism of phosphate groups, nucleic acids, proteins, and lipids . T-AOC represents a highly intuitive embodiment of the antioxidant capacity of the body. In conjunction with SOD, T-AOC actively participates in the process of scavenging free radicals within the body .

Line 426 to 427 has been rephrased and clarified: Generally, the growth performance, immune function, antioxidant capacity, and immune response level of tilapia are influenced by the growing water environment. 

Line 426 to 427 has been rephrased and clarified: The above results indicate that BS and CS can mitigate tilapia growth and immune suppression caused by the increase in COD, TOC, reactive phosphate, nitrite, and ammonia nitrogen contents in the water environment. Additionally, maintaining stable pH and DO levels positively regulate the immune response of tilapia, enhance feed conversion rate, improve growth, and eventually increase the immune status of tilapia. KMPS and THPS can accelerate the degradation of harmful substances in water and improve water quality in a short period. However, due to their strong biocidal properties, prolonged usage of KMPS and THPS may disrupt the stability of the water biota environment, impacting the liver and spleen function of tilapia, there by leading to tissue damage.

Question 19. Line 311 discuss…………..correct 

Answer: The "discuss" has been replaced with "discussion" in line 301 the manuscript .

Question 20. -"At present, the deterioration of water quality caused by environmental pollution has become the main reason for the breeding of aquatic animal diseases"what do you mean?………… this sentence is not correct .

Answer: The sentence has been deleted in the manuscript. 

Question 21. - "Drug prevention can effectively………." what do you mean?………… this sentence is not correct 

Answer: The sentence has been deleted in the manuscript.

Conclusion 

Question 22. Conclusion need more clarification

Answer: The conclusion has been restated and sorted out in line 444 to 451 the manuscript.

Line 444 to 451: In summary, BS and CS can effectively reduce harmful substance levels while improving water quality, growth performance, immune levels, antioxidant capacity, and immune response of tilapia. Moreover, KMPS and THPS can improve not only the water quality but also the immune response and growth performance of tilapia in the short term. However, with prolonged usage, KMPS and THPS can cause damage to water quality and negatively impact tilapia growth. Moreover, this studies have demonstrated that the escalation of deleterious substances in aquatic environments exerts an immunosuppressive impact on tilapia. Conversely, DO has been observed to possess a beneficial regulatory influence on the immune responses of tilapia.

Question 22. -"With the increase of use time, it will not only cause water quality damage, but also damage the liver, spleen and kidney function" what do you mean by "it"

Answer: "It" refers to the KMPS and THPS, and they have been revised in line 447 to 448 the manuscript.

Line 447 to 448: However, with prolonged usage, KMPS and THPS can cause damage to water quality and negatively impact tilapia growth.

---

## [Decision Letter · Decision Letter 1]

17 Aug 2023

Effects of Different Water Quality Regulators on Growth Performance, Immunologic Function, and Domestic Water Quality of GIFT Tilapia

PONE-D-23-10118R1

Dear Dr. Ying-Yi Wei,

We’re pleased to inform you that your manuscript has been judged scientifically suitable for publication and will be formally accepted for publication once it meets all outstanding technical requirements.

Kind regards,

Amel Mohamed El Asely

Academic Editor

PLOS ONE

Additional Editor Comments (optional):

Reviewers' comments:

Reviewer's Responses to Questions

**Comments to the Author**

1. If the authors have adequately addressed your comments raised in a previous round of review and you feel that this manuscript is now acceptable for publication, you may indicate that here to bypass the “Comments to the Author” section, enter your conflict of interest statement in the “Confidential to Editor” section, and submit your "Accept" recommendation.

Reviewer #1: All comments have been addressed

Reviewer #2: All comments have been addressed

2. Is the manuscript technically sound, and do the data support the conclusions?

Reviewer #1: Yes

Reviewer #2: Yes

3. Has the statistical analysis been performed appropriately and rigorously? 

Reviewer #1: Yes

Reviewer #2: Yes

4. Have the authors made all data underlying the findings in their manuscript fully available?

Reviewer #1: Yes

Reviewer #2: Yes

5. Is the manuscript presented in an intelligible fashion and written in standard English?

Reviewer #1: Yes

Reviewer #2: Yes

6. Review Comments to the Author

Reviewer #1: (No Response)

Reviewer #2: accept in its current form...........................................................................

7. PLOS authors have the option to publish the peer review history of their article (what does this mean?). If published, this will include your full peer review and any attached files.

Reviewer #1: No

Reviewer #2: No

---

## [Editor Report · Acceptance letter]

21 Aug 2023

PONE-D-23-10118R1 

Effects of Different Water Quality Regulators on Growth Performance, Immunologic Function, and Domestic Water Quality of GIFT Tilapia 

Dear Dr. Wei:

I'm pleased to inform you that your manuscript has been deemed suitable for publication in PLOS ONE. Congratulations! Your manuscript is now with our production department. 

Kind regards, 

on behalf of

Prof. Amel Mohamed El Asely 

Academic Editor

PLOS ONE